# QUERYING EASILY FLIP-FLOPPED SAMPLES FOR DEEP ACTIVE LEARNING

**Seong Jin Cho**[1,2]**, Gwangsu Kim**[3]**, Junghyun Lee**[4]**, Jinwoo Shin**[4]**, Chang D. Yoo**[1*]
[1]Department of Electrical Engineering, KAIST, [2]Korea Institute of Oriental Medicine,
[3]Department of Statistics, Jeonbuk National University, [4]Kim Jaechul Graduate School of AI, KAIST
{ipcng00,jh_lee00,jinwoos,cd_yoo}@kaist.ac.kr, s88012@jbnu.ac.kr

## ABSTRACT

Active learning, a paradigm within machine learning, aims to select and query unlabeled data to enhance model performance strategically. A crucial selection strategy leverages the model's predictive uncertainty, reflecting the informativeness of a data point. While the sample's distance to the decision boundary intuitively measures predictive uncertainty, its computation becomes intractable for complex decision boundaries formed in multiclass classification tasks. This paper introduces the *least disagree metric* (LDM), the smallest probability of predicted label disagreement. We propose an asymptotically consistent estimator for LDM under mild assumptions. The estimator boasts computational efficiency and straightforward implementation for deep learning models using parameter perturbation. The LDM-based active learning algorithm queries unlabeled data with the smallest LDM, achieving state-of-the-art *overall* performance across various datasets and deep architectures, as demonstrated by the experimental results.

## 1 INTRODUCTION

Machine learning frequently necessitates the laborious annotation of vast and readily available unlabeled datasets. Active learning (AL) (Cohn et al., 1996) addresses this challenge by strategically selecting the most informative unlabeled samples. Within AL algorithms, uncertainty-based sampling (Ash et al., 2020; Zhao et al., 2021b; Woo, 2023) enjoys widespread adoption due to its simplicity and computational efficiency. This approach prioritizes selecting unlabeled samples with the greatest prediction difficulty (Settles, 2009; Yang et al., 2015; Sharma & Bilgic, 2017). The core objective of uncertainty-based sampling lies in quantifying the uncertainty associated with each unlabeled sample for a given predictor. A plethora of uncertainty measures and corresponding AL algorithms have been proposed (Nguyen et al., 2022; Houlsby et al., 2011; Jung et al., 2023); see Appendix A for a more comprehensive overview of the related work. However, many suffer from scalability and interpretability limitations, often relying on heuristic approximations devoid of robust theoretical foundations. This paper presents a novel approach centered on the distance between samples and the decision boundary.

Intuitively, samples closest to the decision boundary are considered most uncertain (Kremer et al., 2014; Ducoffe & Precioso, 2018). Theoretically, selecting unlabeled samples with the smallest margin in binary classification with linear separators demonstrably leads to exponential performance gains compared to random sampling (Balcan et al., 2007; Kpotufe et al., 2022). However, determining the sample-to-decision boundary distance is computationally infeasible in most cases, particularly for multiclass predictors based on deep neural networks. While various approximate measures have been proposed to identify such closest samples (Ducoffe & Precioso, 2018; Moosavi-Dezfooli et al., 2016; Mickisch et al., 2020), most lack substantial justification.

This paper proposes a paradigm shift in the realm of closeness measures. We introduce the least disagree metric (LDM), which quantifies a sample's closeness to the decision boundary by deviating from the conventional Euclidean-based distance. Samples identified as closest based on this metric harbor the highest prediction uncertainty and represent the most informative data points. Our work presents the following key contributions:

---

[*]Corresponding author

- Definition of the LDM as a measure of a sample's closeness to the decision boundary.
- Introduction of an asymptotically consistent LDM estimator under mild assumptions and a straightforward algorithm for its empirical evaluation, inspired by theoretical analysis. This algorithm leverages Gaussian perturbation centered around the hypothesis learned by stochastic gradient descent (SGD).
- proposition of an LDM-based AL algorithm (LDM-S) showcasing state-of-the-art overall performance across six benchmark image datasets and three OpenML datasets, tested over various deep architectures.

## 2 LEAST DISAGREE METRIC (LDM)

This section formally defines the concept of the *least disagree metric* (LDM) and proposes an asymptotically consistent estimator for its calculation. Drawing inspiration from a Bayesian perspective, we present a practical algorithm for the empirical evaluation of the LDM.

### 2.1 DEFINITION OF LDM

Let $\mathcal{X}$ and $\mathcal{Y}$ be the instance and label spaces, respectively, with $|\mathcal{Y}| \geq 2$, and $\mathcal{H}$ be the hypothesis space, encompassing all possible functions $h : \mathcal{X} \to \mathcal{Y}$. Let $\mathcal{D}$ be the joint distribution over $\mathcal{X} \times \mathcal{Y}$, and $\mathcal{D}_{\mathcal{X}}$ be the marginal distribution over instances only. The LDM is inspired by the *disagree metric* introduced by Hanneke (Hanneke, 2014). The disagree metric between two hypotheses $h_1$ and $h_2$ is defined as follows:

$$\rho(h_1, h_2) := \mathbb{P}_{X \sim \mathcal{D}_{\mathcal{X}}}[h_1(X) \neq h_2(X)] \tag{1}$$

where $\mathbb{P}_{X \sim \mathcal{D}_{\mathcal{X}}}$ is the probability measure on $\mathcal{X}$ induced by $\mathcal{D}_{\mathcal{X}}$. For a given hypothesis $g \in \mathcal{H}$ and $\boldsymbol{x}_0 \in \mathcal{X}$, let $\mathcal{H}^{g,\boldsymbol{x}_0} := \{h \in \mathcal{H} \mid h(\boldsymbol{x}_0) \neq g(\boldsymbol{x}_0)\}$ be the set of hypotheses disagreeing with $g$ in their prediction for $\boldsymbol{x}_0$. Based on the above set and disagree metric, the LDM of a sample to a given hypothesis is defined as follows:

**Definition 1.** For given $g \in \mathcal{H}$ and $\boldsymbol{x}_0 \in \mathcal{X}$, the **least disagree metric (LDM)** is defined as

$$L(g, \boldsymbol{x}_0) := \inf_{h \in \mathcal{H}^{g,\boldsymbol{x}_0}} \rho(h, g). \tag{2}$$

Throughout the paper, we assume all the hypotheses belong to a parametric family, and $h, g$ are identified by $\boldsymbol{w}, \boldsymbol{v} \in \mathbb{R}^p$, respectively.

To illustrate the LDM concept, we consider a two-dimensional binary classification with a set of linear classifiers defined as

$$\mathcal{H} = \{h : h(\boldsymbol{x}) = \text{sign}(\boldsymbol{x}^\mathsf{T}\boldsymbol{w}), \boldsymbol{w} \in \mathbb{R}^2\}$$

where $\boldsymbol{x}$ is uniformly distributed on $\mathcal{X} = \{\boldsymbol{x} : \|\boldsymbol{x}\| \leq 1\} \subset \mathbb{R}^2$. Let $h_\theta$ be a hypothesis corresponding to a decision boundary forming an angle $\theta \in (-\pi, \pi)$ (in radians) with the decision boundary of $g$, then we have that $\rho(h_\theta, g) = \frac{|\theta|}{\pi}$. For given $g$ and $\boldsymbol{x}_0$, let $\theta_0 \in (0, \frac{\pi}{2})$ be the angle between the decision boundary of $g$ and the line passing through $\boldsymbol{x}_0$, then $\mathcal{H}^{g,\boldsymbol{x}_0} = \{h_\theta \mid \theta \in (-\pi, -\pi+\theta_0) \cup (\theta_0, \pi)\}$; see Figure 1. Thus, $L(g, \boldsymbol{x}_0) = \inf_{h_\theta \in \mathcal{H}^{g,\boldsymbol{x}_0}} \rho(h_\theta, g) = \frac{|\theta_0|}{\pi}$.

Figure 1: An example of LDM of $\boldsymbol{x}_0$ for given $g$ in binary classification with the linear classifier. Here $\boldsymbol{x}$ is uniformly distributed on $\mathcal{X} \subset \mathbb{R}^2$. The $h_\theta$ disagrees with $g$ for $\boldsymbol{x}_0$ when $\theta < -\pi+\theta_0$ or $\theta_0 < \theta$, thus $L(g, \boldsymbol{x}_0) = \inf_{h_\theta \in \mathcal{H}^{g,\boldsymbol{x}_0}} \rho(h_\theta, g) = \frac{|\theta_0|}{\pi}$.

Conceptually, a sample with a smaller LDM signifies that its predicted label can be more easily flip-flopped by even a minimal perturbation to the decision boundary. Suppose we draw a hypothesis $h$ with its parameters sampled from a Gaussian distribution centered at the parameters of $g$ (i.e., $\boldsymbol{w} \sim \mathcal{N}(\boldsymbol{v}, \mathbf{I}\sigma^2)$), then it is expected that

$$L(g, \boldsymbol{x}_1) < L(g, \boldsymbol{x}_2) \Leftrightarrow \mathbb{P}_{\boldsymbol{w}}[h(\boldsymbol{x}_1) \neq g(\boldsymbol{x}_1)] > \mathbb{P}_{\boldsymbol{w}}[h(\boldsymbol{x}_2) \neq g(\boldsymbol{x}_2)].$$

The sample with the smallest LDM is the most uncertain and, thus, most informative. The theoretical justification for this intuition is established specifically for two-dimensional binary classification with linear separators. Additionally, empirical verification is provided on benchmark datasets utilizing deep networks (see Appendix C.1).

## 2.2 AN ASYMPTOTICALLY CONSISTENT ESTIMATOR OF LDM

In most cases, LDM is not computable for the following two reasons: 1) $\rho$ is generally intractable, especially when $\mathcal{D}_{\mathcal{X}}$ and $h$ are both complicated, e.g., neural networks over real-world image datasets, and 2) one needs to take an infimum over $\mathcal{H}$, which is usually an infinite set.

To address these issues, we propose an estimator for the LDM based on the following two approximations: 1) $\mathbb{P}$ in the definition of $\rho$ is replaced by an empirical probability based on $M$ samples (Monte-Carlo method), and 2) $\mathcal{H}$ is replaced by a finite hypothesis set $\mathcal{H}_N$ of cardinality $N$. Our estimator, denoted by $L_{N,M}(g, \boldsymbol{x}_0)$, is defined as follows:

$$L_{N,M}(g, \boldsymbol{x}_0) := \inf_{h \in \mathcal{H}_N^{g, \boldsymbol{x}_0}} \left\{ \rho_M(h, g) \triangleq \frac{1}{M} \sum_{i=1}^{M} \mathbb{I}\big[h(X_i) \neq g(X_i)\big] \right\}, \tag{3}$$

where $\mathcal{H}_N^{g, \boldsymbol{x}_0} := \{h \in \mathcal{H}_N \mid h \in \mathcal{H}^{g, \boldsymbol{x}_0}\}$, $\mathbb{I}[\cdot]$ is an indicator function, and $X_1, \ldots, X_M \overset{i.i.d.}{\sim} \mathcal{D}_{\mathcal{X}}$. Here, $M$ is the number of Monte Carlo samples for approximating $\rho$, and $N$ is the number of sampled hypotheses for approximating $L$.

A critical property of an estimator is (asymptotic) consistency, i.e., in our case, we want the LDM estimator to converge in probability to the true LDM as $M$ and $N$ increase indefinitely.

We start with two assumptions on the hypothesis space $\mathcal{H}$ and the disagree metric $\rho$:

**Assumption 1.** *$\mathcal{H}$ is a Polish space with metric $d_{\mathcal{H}}(\cdot, \cdot)$.*

This assumption allows us to avoid any complications[1] that may arise from uncountability, especially as we will consider a probability measure over $\mathcal{H}$; see Chapter 1.1 of van der Vaart & Wellner (1996).

**Assumption 2.** *$\rho(\cdot, \cdot)$ is $B$-Lipschitz for some $B > 0$, i.e., $\rho(h, g) \leq B d_{\mathcal{H}}(h, g)$, $\forall h, g \in \mathcal{H}$.*

If $\rho$ is not Lipschitz, then the disagree metric may behave arbitrarily regardless of whether the hypotheses are "close" or not. This can occur in specific (arguably not so realistic) corner cases, such as when $\mathcal{D}_{\mathcal{X}}$ is a mixture of Dirac measures.

From hereon and forth, we fix some $g \in \mathcal{D}$ and $\boldsymbol{x}_0 \in \mathcal{X}$.

The following assumption intuitively states that $\mathcal{H}_N$ is more likely to cover regions of $\mathcal{H}$ whose LDM estimator is $\varepsilon$-close to the true LDM, as the number of sampled hypotheses $N$ increases.

**Assumption 3** (Coverage Assumption). *There exist two deterministic functions $\alpha : \mathbb{N} \times \mathbb{R}_{\geq 0} \to \mathbb{R}_{\geq 0}$ and $\beta : \mathbb{N} \to \mathbb{R}_{>0}$ that satisfies $\lim_{N \to \infty} \alpha(N, \varepsilon) = \lim_{N \to \infty} \beta(N) = 0$ for any $\varepsilon \in (0, 1)$, and*

$$\mathbb{P}\left[\inf_{\substack{h^* \in \mathcal{H}^{g, \boldsymbol{x}_0} \\ \rho(h^*, g) - L(g, \boldsymbol{x}_0) \leq \varepsilon}} \min_{h \in \mathcal{H}_N^{g, \boldsymbol{x}_0}} d_{\mathcal{H}}(h^*, h) \leq \frac{1}{B}\alpha(N, \varepsilon)\right] \geq 1 - \beta(N), \quad \forall \varepsilon \in (0, 1), \tag{4}$$

*where we recall that $\mathcal{H}^{g, \boldsymbol{x}_0} = \{h \in \mathcal{H} \mid h(\boldsymbol{x}_0) \neq g(\boldsymbol{x}_0)\}$.*

Here, we implicitly assume that there is a randomized procedure that outputs a sequence of finite sets $\mathcal{H}_1 \subseteq \mathcal{H}_2 \subseteq \cdots \subset \mathcal{H}$. $\alpha(N, \varepsilon)$ is the rate describing the optimality of approximating $\mathcal{H}$ using $\mathcal{H}_N$ in that how close is the $\varepsilon$-optimal solution is, and $1 - \beta(N)$ is the confidence level that converges to 1.

In the following theorem, whose proof is deferred to Appendix B, we show that our proposed estimator, $L_{N,M}$, is asymptotically consistent:

**Theorem 1.** *Let $g \in \mathcal{H}$, $\boldsymbol{x}_0 \in \mathcal{X}$, and $\delta > 0$ be arbitrary. Under Assumption 1, 2, and 3, with $M > \frac{8}{\delta^2} \log(CN)$, we have that for any $\varepsilon \in (0, 1)$,*

$$\mathbb{P}\left[|L_{N,M}(g, \boldsymbol{x}_0) - L(g, \boldsymbol{x}_0)| \leq 2\delta + \alpha(N, \varepsilon) + \varepsilon\right] \geq 1 - \frac{1}{CN} - \beta(N).$$

*Furthermore, as $\min(M, N) \to \infty$ with[2] $M = \omega\left(\log(CN)\right)$, we have $L_{N,M}(g, \boldsymbol{x}_0) \xrightarrow{\mathbb{P}} L(g, \boldsymbol{x}_0)$.*

---

[1]The usual measurability and useful properties may not hold for non-separable spaces (e.g., Skorohod space).
[2]For the asymptotic analyses, we write $f(n) = \omega(g(n))$ if $\lim_{n \to \infty} \frac{f(n)}{g(n)} = \infty$.

For our asymptotic guarantee, we require $M = \omega(\log(CN))$, while the guarantee holds w.p. at least $1 - \frac{1}{CN} - \beta(N)$. For instance, when $\beta(N)$ scales as $1/N$, the above implies that when $N$ scales exponentially (in dimension or some other quantity), the guarantee holds with overwhelming probability while $M$ only needs to be at least scaling linearly, i.e., $M$ needs not be too large.

One important consequence is that the ordering of the empirical LDM is preserved in probability:

**Corollary 1.** *Assume that $L(g, \boldsymbol{x}_i) < L(g, \boldsymbol{x}_j)$. Under the same assumptions as Theorem 1, the following holds: for any $\varepsilon > 0$,*

$$\lim_{\min(M,N) \to \infty} \mathbb{P}\left[L_{N,M}(g, \boldsymbol{x}_i) > L_{N,M}(g, \boldsymbol{x}_j) + \varepsilon\right] = 0,$$

*where we again require that $M = \omega\left(\log(CN)\right)$.*

### 2.3 EMPIRICAL EVALUATION OF LDM

**Motivation.** LDM estimator requires Assumption 3 to hold for asymptotic consistency. This ensures that with a sufficiently large number of samples ($N \to \infty$), the constructed hypothesis set has a high probability of "covering" the true LDM-yielding hypothesis. We employ Gaussian sampling around the target hypothesis $g$ to construct a finite collection of hypotheses $\mathcal{H}_N$ to achieve this coverage. Appendix C.3 formally demonstrates that this approach satisfies Assumption 3 in the context of 2D binary classification with linear classifiers.

**Remark 1.** *It is known that SGD performs Bayesian inference (Mandt et al., 2017; Chaudhari & Soatto, 2018; Mingard et al., 2021), i.e., $g$ can be thought of as a sample from a posterior distribution. Combined with the Bernstein-von Mises theorem (Hjort et al., 2010), which states that the posterior of a parametric model converges to a normal distribution under mild conditions, Gaussian sampling around $g$ can be thought of as sampling from a posterior distribution, in an informal sense.*

**Algorithm Details.** Algorithm 1 provides a method for empirically evaluating the LDM of $\boldsymbol{x}$ for given $g$. When constructing the hypothesis set, we utilize a set of variances $\{\sigma_k^2\}_{k=1}^K$ such that $\sigma_k < \sigma_{k+1}$. The choice is motivated by two key considerations. If $\sigma^2$ is too small, the sampled $h$ is unlikely to satisfy $h(\boldsymbol{x}) \neq g(\boldsymbol{x})$, which is required for LDM estimation. Conversely, excessively large $\sigma^2$ can lead to $h$ too far from $g$ when the true LDM is close to 0. Both observations are based on the intuition that larger $\sigma^2$ implies a greater $\rho(h, g)$. Appendix C.2 formally proves that for 2D binary classification with linear classifiers, $\mathbb{E}_{\boldsymbol{w}}[\rho_M(h, g)]$ is monotonically increasing with respect to $\sigma^2$. We also provide empirical evidence supporting this observation in more realistic scenarios. The algorithm proceeds as follows: For each $k$, sample $h$ with $\boldsymbol{w} \sim \mathcal{N}(\boldsymbol{v}, \mathbf{I}\sigma_k^2)$. If $h(\boldsymbol{x}) \neq g(\boldsymbol{x})$, update $L_{\boldsymbol{x}}$ as $\min\{L_{\boldsymbol{x}}, \rho_M(h, g)\}$. If $L_{\boldsymbol{x}}$ remains unchanged for $s$ consecutive iterations, move on to $k+1$. Continue iterating while $k < K$.

**Small $s$ is Sufficient.** Choosing an appropriate $s$ remains an open question. While a small $s$ suffices for accurate LDM estimation in the context of binary classification with the linear classifier depicted in Figure 1, deep architectures might require a larger $s$ to ensure convergence of the estimated LDM. However, employing a large $s$ can be computationally expensive due to the significantly increased runtime associated with sampling a larger number of hypotheses. Fortunately, our empirical observations across various $s$ reveal that the rank order of LDM across different samples remains consistent. This is quantified by a high rank-correlation coefficient close to 1. Since the LDM-based active learning algorithm described in Section 3 leverages only the relative ordering of LDM for sample selection, a small $s$ is sufficient for practical purposes. Detailed experimental results and further discussions regarding the impact of the $s$ are provided in Appendix F.1.

## 3 LDM-BASED ACTIVE LEARNING

This section introduces the LDM-based batch sampling algorithm (LDM-S) for pool-based active learning. In pool-based active learning, we are given a set of unlabeled samples denoted by $\mathcal{U}$. The goal is to select a batch of $q$ informative samples for querying from a randomly sampled pool $\mathcal{P} \subset \mathcal{U}$ of size $m$.

**Algorithm 1** Empirical Evaluation of LDM

**Input:**
$\boldsymbol{x}$: target sample
$g$: target hypothesis parameterized by $\boldsymbol{v}$
$M$: number of samples for approximation
$\{\sigma_k^2\}_{k=1}^K$: set of variances
$s$: stop condition for parameter sampling

$L_{\boldsymbol{x}} = 1$
**for** $k = 1$ **to** $K$ **do**
  $c = 0$
  **while** $c < s$ **do**
    Sample $h$ with $\boldsymbol{w} \sim \mathcal{N}(\boldsymbol{v}, \mathbf{I}\sigma_k^2)$
    $c = c + 1$
    **if** $h(\boldsymbol{x}) \neq g(\boldsymbol{x})$ and $L_{\boldsymbol{x}} > \rho_M(h, g)$ **then**
      $L_{\boldsymbol{x}} \leftarrow \rho_M(h, g)$
      $c = 0$
    **end if**
  **end while**
**end for**
**return:** $L_{\boldsymbol{x}}$

**Algorithm 2** Active Learning with LDM-S

**Input:**
$\mathcal{L}_0, \mathcal{U}_0$ : Initial labeled and unlabeled samples
$m, q$ : pool and query size
$T$ : number of acquisition steps

**for** $t = 0$ **to** $T - 1$ **do**
  Obtain $\boldsymbol{v}$ by training on $\mathcal{L}_t$
  Randomly sample $\mathcal{P} \subset \mathcal{U}_t$ with $|\mathcal{P}| = m$
  Evaluate $L_{\boldsymbol{x}}$ of $\boldsymbol{x} \in \mathcal{P}$ for $\boldsymbol{v}$ by Algorithm 1
  Compute $\gamma_{\boldsymbol{x}}$ using Eqn. 5
  $\mathcal{Q}_1 \leftarrow \{\boldsymbol{x}_1\}$ where $\boldsymbol{x}_1 = \arg\min_{\boldsymbol{x} \in \mathcal{P}} L_{\boldsymbol{x}}$
  **for** $n = 2$ **to** $q$ **do**
    $p_{\boldsymbol{x}} = \gamma_{\boldsymbol{x}} * \min_{\boldsymbol{x}' \in \mathcal{Q}_{n-1}} d_{\cos}(\boldsymbol{z}_{\boldsymbol{x}}, \boldsymbol{z}_{\boldsymbol{x}'})$
    Sample $\boldsymbol{x}_n \in \mathcal{P}$ w.p. $\mathbb{P}(\boldsymbol{x}_n) = \frac{p_{\boldsymbol{x}_n}^2}{\sum_{\boldsymbol{x}_j \in \mathcal{P}} p_{\boldsymbol{x}_j}^2}$
    $\mathcal{Q}_n \leftarrow \mathcal{Q}_{n-1} \cup \{\boldsymbol{x}_n\}$
  **end for**
  $\mathcal{L}_{t+1} \leftarrow \mathcal{L}_t \cup \{(\boldsymbol{x}_i, y_i)\}_{\boldsymbol{x}_i \in \mathcal{Q}_q}, \mathcal{U}_{t+1} \leftarrow \mathcal{U}_t \setminus \mathcal{Q}_q$
**end for**

## 3.1 LDM-SEEDING

A naïve approach for LDM-based active learning would be to select the $q$ samples with the smallest LDMs (most uncertain). The effectiveness of LDM-based active learning is shown in Figure 2 of Section 4.1. However, Figure 3 of Section 4.2 demonstrates that this strategy may not yield optimal performance since selecting samples solely based on LDM can lead to significant overlap in the information content of the chosen samples. This issue, referred to as *sampling bias*, is a common challenge in uncertainty-based active learning approaches (Dasgupta, 2011). Several methods have been proposed to mitigate sampling bias by incorporating diversity into the selection process. These methods include $k$-means++ seeding (Ash et al., 2020), submodular maximization (Wei et al., 2015), clustering (Citovsky et al., 2021; Yang et al., 2021), joint mutual information (Kirsch et al., 2019).

This paper adopts a modified version of the $k$-means++ seeding algorithm (Arthur & Vassilvitskii, 2007) to promote batch diversity without introducing additional hyperparameters (Ash et al., 2020). Intuitively, $k$-means++ seeding iteratively selects centroids by prioritizing points further away from previously chosen centroids. This method naturally tends to select a diverse set of points. LDM-Seeding employs the cosine distance between the last layer features of the deep network as the distance metric. This choice is motivated by the fact that the perturbation is applied to the weights of the last layer and that feature scaling does not affect the final prediction.

LDM-Seeding balances selecting samples with the smallest LDMs while achieving diversity. This is achieved through a modified exponentially decaying weighting scheme inspired by the EXP3 algorithm (Auer et al., 2002). This type of weighting scheme has proven successful in various active learning (Beygelzimer et al., 2009; Ganti & Gray, 2012; Kim & Yoo, 2022). To balance the competing goals, $\mathcal{P}$ is partitioned into $\mathcal{P}_q$ and $\mathcal{P}_c$ where $\mathcal{P}_q$ contains $q$ samples with the smallest LDMs and $\mathcal{P}_c = \mathcal{P} \setminus \mathcal{P}_q$. The total weights assigned to $\mathcal{P}_q$ and $\mathcal{P}_c$ are set to be equal. For each sample $\boldsymbol{x} \in \mathcal{P}_i$, let $L_q = \max_{\boldsymbol{x} \in \mathcal{P}_q} L_{\boldsymbol{x}}$, then the weight $\gamma_{\boldsymbol{x}}$ is calculated using the following equation:

$$\gamma_{\boldsymbol{x}} = \frac{e^{-\eta_{\boldsymbol{x}}}}{\sum_{\boldsymbol{x} \in \mathcal{P}_i} e^{-\eta_{\boldsymbol{x}}}}, \quad \eta_{\boldsymbol{x}} = \frac{(L_{\boldsymbol{x}} - L_q)_+}{L_q} \tag{5}$$

where $i \in \{q, c\}$ denotes the partition index and $(\cdot)_+ = \max\{0, \cdot\}$.

LDM-Seeding starts by selecting an unlabeled sample with the smallest LDM from $\mathcal{P}$. Subsequently, it employs the following probability distribution to choose the next distant unlabeled sample:

$$\mathbb{P}(\boldsymbol{x}) = \frac{p_{\boldsymbol{x}}^2}{\sum_{\boldsymbol{x}_j \in \mathcal{P}} p_{\boldsymbol{x}_j}^2}, \quad p_{\boldsymbol{x}} = \gamma_{\boldsymbol{x}} * \min_{\boldsymbol{x}' \in \mathcal{Q}} d_{\cos}(\boldsymbol{z}_{\boldsymbol{x}}, \boldsymbol{z}_{\boldsymbol{x}'}) \tag{6}$$

where $Q$ is the set of selected samples, and $\boldsymbol{z}_{\boldsymbol{x}}, \boldsymbol{z}_{\boldsymbol{x}'}$ are the features of $\boldsymbol{x}, \boldsymbol{x}'$, respectively. This probability distribution explicitly incorporates both LDM and diversity.

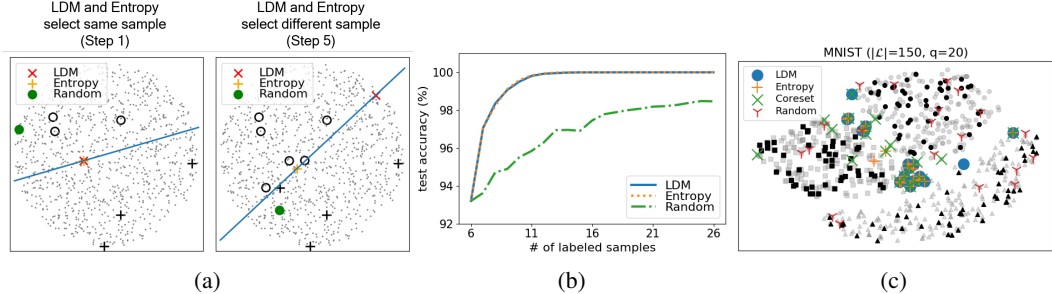

(a)  (b)  (c)

Figure 2: The comparison of selecting sample(s). The black crosses and circles are labeled, and the gray dots are unlabeled samples. (a) Selected samples by LDM-based, entropy-based, and random sampling in binary classification with the linear classifier. (b) The test accuracy with respect to the number of labeled samples. (c) The t-SNE plot of selected batch samples in 3-class classification with a deep network on MNIST dataset.

## 3.2 LDM-S: Active Learning with LDM-Seeding

We now introduce LDM-S in Algorithm 2, the LDM-based seeding algorithm for active learning. Let $\mathcal{L}_t$ and $\mathcal{U}_t$ be the set of labeled and unlabeled samples at step $t$, respectively. For each step $t$, the given parameter $v$ is obtained by training on $\mathcal{L}_t$, and the pool data $\mathcal{P} \subset \mathcal{U}_t$ of size $m$ is drawn uniformly at random. Then for each $x \in \mathcal{P}$, $L_x$ for given $v$ and $\gamma_x$ are evaluated by Algorithm 1 and Eqn. 5, respectively. The set of selected unlabeled samples, $\mathcal{Q}_1$, is initialized as $\{x_1\}$ where $x_1 = \arg\min_{x \in \mathcal{P}} L_x$. For $n = 2, \ldots, q$, the algorithm samples $x_n \in \mathcal{P}$ with probability $\mathbb{P}(x)$ in Eqn. 6 and appends it to $\mathcal{Q}_n$. Lastly, the algorithm queries the label $y_i$ of each $x_i \in \mathcal{Q}_q$, and the algorithm continues until $t = T - 1$.

## 4 Experiments

This section presents the empirical evaluation of the proposed LDM-based active learning algorithm. We compare its performance against various uncertainty-based active learning algorithms on diverse datasets: 1) three OpenML (OML) datasets (#6 (Frey & Slate, 1991), #156 (Vanschoren et al., 2014), and #44135 (Fanty & Cole, 1990)); 2) six benchmark image datasets (MNIST (Lecun et al., 1998), CIFAR10 (Krizhevsky, 2009), SVHN (Netzer et al., 2011), CIFAR100 (Krizhevsky, 2009), Tiny ImageNet (Le & Yang, 2015), FOOD101 (Bossard et al., 2014)), and ImageNet (Russakovsky et al., 2015). We employ various deep learning architectures: MLP, S-CNN, K-CNN (Chollet et al., 2015), Wide-ResNet (WRN-16-8; Zagoruyko & Komodakis (2016)), and ResNet-18 (He et al., 2016). All results represent the average performance over 5 repetitions (3 for ImageNet). Detailed descriptions of the experimental settings are provided in Appendix D. The source code is available on the authors' GitHub repository[3].

## 4.1 Effectiveness of LDM in Selecting (Batched) Uncertain Samples

To gain insights into the functionality of LDM, we conducted a controlled active learning experiment under a scenario where the true LDM is readily measurable. We employed a two-dimensional binary classification with $\mathcal{H} = \{h : h_w(x) = \mathbb{I}[\ell_w(x) > 0.5]\}$ where $\ell_w(x) := 1/(1 + e^{-(x^\mathsf{T} w)})$, $w \in \mathbb{R}^2$ is the parameter of $h$, and $x$ is uniformly distributed on $\mathcal{X} = \{x : \|x\| \leq 1\} \subset \mathbb{R}^2$. Three initial labeled samples are randomly chosen from each class. For each step, one sample is queried. LDM-based active learning (LDM) is compared with entropy-based uncertainty sampling (Entropy) and random sampling (Random). Figure 2a shows the selected samples by each algorithm. Black crosses and circles denote labeled samples, while gray dots represent unlabeled samples. Both LDM and Entropy select samples close to the decision boundary. Figure 2b shows the average test accuracy over 100 repetitions for each algorithm, plotted against the number of labeled samples. LDM performs similarly to Entropy, both significantly outperforming Random. This observation aligns with the

---

[3]https://github.com/ipcng00/LDM-S

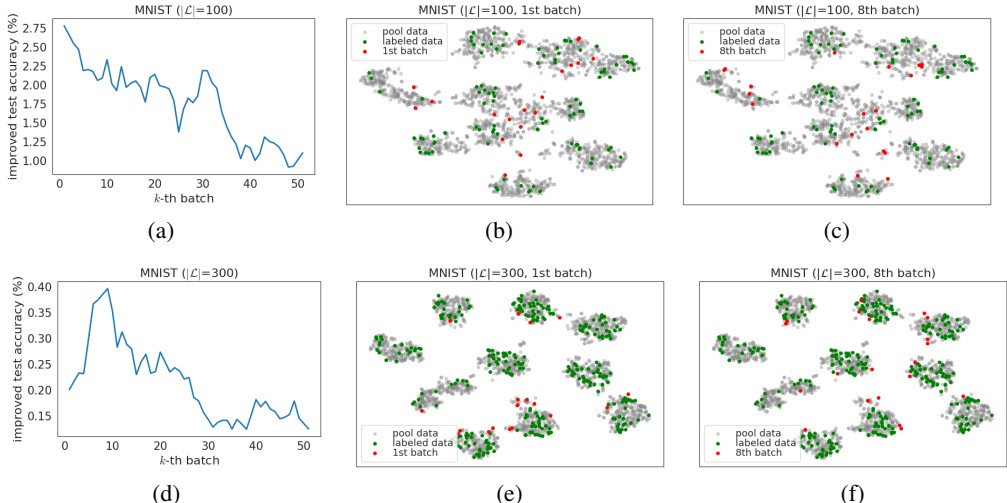

Figure 3: The improved test accuracy by labeling the $k^{\text{th}}$ batch of size $q$ from pool data sorted in ascending order of LDM when the number of labeled samples is 100 (a) or 300 (d), and t-SNE plots of the first and eighth batches for each case (b-c, e-f) on MNIST.

theoretical understanding that entropy is a strongly decreasing function of the distance to the decision boundary in binary classification. Consequently, both LDM and entropy-based algorithms effectively select the sample closest to the decision boundary, mirroring the behavior of the well-established margin-based algorithm. Furthermore, we compared the batch samples chosen by LDM, Entropy, Coreset, and Random in a 3-class classification task using a deep network on the MNIST dataset. Figure 2c shows the t-SNE plot of the selected samples. Gray and black points represent pool data and labeled samples, respectively. The results show that LDM and Entropy select samples close to the decision boundaries, similar to the two-dimensional case. Coreset, on the other hand, selects a more diverse set of samples. For further analysis, the authors conducted additional experiments: 1) investigating the impact of varying batch sizes (Appendix F.3); 2) Evaluating the effectiveness of LDM without using it for seeding (Appendix F.2); 3) Comparing the performance of LDM with other uncertainty measures in the context of seeding algorithm. (Appendix G.1). These additional experiments further substantiate the effectiveness of the newly introduced LDM for active learning.

## 4.2 NECESSITY OF PURSUING DIVERSITY IN LDM-S

Here, we investigate the potential drawbacks of *solely* relying on the smallest LDM values for sample selection, thus why we need to consider diversity. We query the $k^{\text{th}}$ batch of size $q = 20$ from MNIST pool data sorted in ascending order of their LDM values and compare the improvements in test accuracy. Figure 3a shows that selecting samples with the smallest LDMs initially leads to the best performance, but Figure 3d shows that prioritizing the smallest LDMs may not lead to the optimal performance. To understand this difference, we visualize the distribution of the selected samples using t-SNE plots (van der Maaten & Hinton, 2008) for each case. Figure 3b–3c show the t-SNE plots for the first and eighth batches, respectively, when 100 samples are labeled. Both plots display a well-spread distribution of selected samples, implying minimal overlap in information content between subsequent batches. Figure 3e–3f show the t-SNE plots for the first and eighth batches, respectively, when 300 samples are labeled. The samples exhibit significant overlap in the first batch, indicating redundant information. In contrast, the eighth batch, comprising samples with larger LDMs, demonstrates a more diverse spread. Based on these observations, we conclude that diversity within the selected samples is crucial in achieving optimal performance. Even in LDM-based selection, incorporating diversity considerations during batch selection becomes essential. Appendix C.4 delves deeper into this phenomenon through a geometrical interpretation of 2D binary classification with linear classifiers.

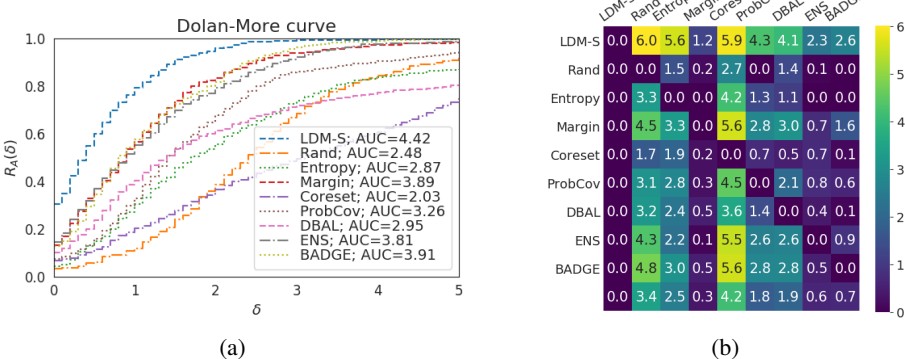

(a)          (b)

Figure 4: The performance comparison across datasets (a) Dolan-Moré plot among the algorithms across all experiments. AUC is the area under the curve. (b) The pairwise penalty matrix over all experiments. Element $P_{i,j}$ corresponds roughly to the number of times algorithm $i$ outperforms algorithm $j$. Column-wise averages at the bottom show overall performance (lower is better).

## 4.3 COMPARING LDM-S TO BASELINE ALGORITHMS

We now compare the performance of the proposed LDM-S with various baseline AL algorithms.

**Baseline algorithms** Each baseline algorithm is denoted as follows: 'Rand': random sampling, 'Entropy': entropy-based uncertainty sampling (Shannon, 1948), 'Margin': margin-based sampling (Roth & Small, 2006), 'Coreset': core-set selection (Sener & Savarese, 2018), 'ProbCov': maximizing probability coverage (Yehuda et al., 2022), 'DBAL': MC-dropout sampling with BALD (Gal et al., 2017), 'ENS': ensemble method with variation ratio (Beluch et al., 2018), 'BADGE': batch active learning by diverse gradient embeddings (Ash et al., 2020), and 'BAIT': batch active learning via information metrics (Ash et al., 2021). We use 100 forward passes for MC-dropout in DBAL and an ensemble of 5 identical but randomly initialized networks for ENS. DBAL can not be conducted on ResNet-18, and BAIT can not be conducted with our resources on CIFAR100, Tiny ImageNet, FOOD101, and ImageNet due to their significantly higher runtime requirements. We set $s = 10$, $M$ to be the same size as the pool size, and $\sigma_k = 10^{\beta k - 5}$ for LDM-S where $\beta = 0.1$ and $k \in \{1, 2, \cdots, 51\}$ in convenient (see Appendix F.4).

**Performance comparison *across* datasets** The performance profile (Dolan & Moré, 2002) and penalty matrix (Ash et al., 2020) are utilized for comprehensive comparisons across all datasets. The details of the performance profile and penalty matrix are described in Appendix E. Figure 4a shows the performance profile of all algorithms with respect to $\delta$. LDM-S consistently maintains the highest $R_A(\delta)$ across all considered $\delta$ values. Notably, $R_{\text{LDM-S}}(0) = 35\%$, significantly exceeding the values of other algorithms (all below 15%). Figure 4b further supports LDM-S's superiority. In the first row, LDM-S outperforms all the other algorithms, often with statistically significant margins (Recall that the largest entry of the penalty matrix is 9, the total number of datasets). Similarly, the first column shows that no other algorithm consistently outperforms LDM-S across all runs. Finally, the bottom row confirms that LDM-S achieves *best* performance.

**Performance comparison *per* datatset** Table 1 presents the mean and standard deviation of performance differences (relative to Random) averaged over repetitions and steps. Positive values indicate better performance than Random, and asterisks (*) denote statistically significant improvements based on paired t-tests. We observe that LDM-S consistently performs best or is comparable to other algorithms across all datasets. In contrast, the effectiveness of other algorithms varies significantly depending on the dataset. For instance, Entropy and Coreset underperform on OpenML, MNIST, CIFAR10, and SVHN compared to LDM-S. Margin performs well on smaller datasets but deteriorates on larger ones. ENS, ProbCov, DBAL, BADGE, and BAIT generally underperform compared to LDM-S. Detailed test accuracy values for each dataset are provided in Appendix G.3.

Table 1: The mean $\pm$ standard deviation of the repetition-wise averaged performance (test accuracy) differences (%), relative to Random, over the entire steps. The positive value indicates higher performance than Random, and the asterisk (*) indicates that $p < 0.05$ in paired sample $t$-test between LDM-S and others. (**bold+underlined**: best performance, **bold**: second-best performance)

| | LDM-S | Entropy | Margin | Coreset | ProbCov | DBAL | ENS | BADGE | BAIT |
|---|---|---|---|---|---|---|---|---|---|
| OML#6 | **4.76**±0.36 | -2.46±0.70* | 4.11±0.24 | 1.14±0.33* | 0.19±0.40* | 0.10±0.22* | **4.43**±0.31 | 2.98±0.28* | 2.57±0.30* |
| OML#156 | **1.18**±0.32 | -1.29±0.51* | 0.64±0.39* | -19.53±3.32* | -0.08±0.51* | -14.40±1.08* | 0.61±0.35* | **1.06**±0.39 | -7.66±1.12* |
| OML#44135 | **4.36**±0.27 | 1.84±0.36* | **4.11**±0.32 | 0.27±0.68* | 3.55±0.28* | 1.61±0.45* | 2.47±0.37* | 2.48±0.44* | 2.55±0.41* |
| MNIST | **3.33**±0.43 | 2.36±0.84* | 3.01±0.45 | -0.04±1.23* | 2.92±0.41 | 1.68±0.80* | 2.98±0.36 | 3.01±0.45 | **3.37**±0.43 |
| CIFAR10 | **1.34**±0.19 | 0.00±0.21* | 0.43±0.27* | -3.71±0.56* | 0.04±0.37* | -0.15±0.31* | 0.58±0.28* | **0.90**±0.21* | 0.60±0.13* |
| SVHN | **2.53**±0.22 | 1.52±0.19* | 1.98±0.18* | -1.66±0.51* | 0.88±0.14* | **2.46**±0.21 | 2.08±0.22* | 2.18±0.23* | 2.18±0.23* |
| CIFAR100 | **0.98**±0.44 | 0.37±0.60 | 0.57±0.45 | **0.89**±0.49 | 0.74±0.60 | 0.55±0.77 | 0.03±0.41* | 0.64±0.48 | - |
| T. ImageNet | **0.55**±0.16 | -0.61±0.28* | 0.28±0.29 | -0.20±0.46* | **0.45**±0.23 | 0.27±0.19* | -0.15±0.35* | 0.12±0.40* | - |
| FOOD101 | **1.27**±0.34 | -0.86±0.20* | 0.34±0.25* | **1.30**±0.16 | 0.50±0.22* | 1.18±0.35 | -0.15±0.46* | 0.71±0.43* | - |
| ImageNet | **0.96**±0.23 | -0.21±0.58* | 0.14±0.54 | -0.62±0.20* | 0.30±0.22* | - | 0.57±0.21 | **0.71**±0.81 | - |

Table 2: The mean of runtime (min) for each algorithm and each dataset. We observe that LDM-S operates as fast as Entropy on almost all datasets.

| | LDM-S | Entropy | Margin | Coreset | ProbCov | DBAL | ENS | BADGE | BAIT |
|---|---|---|---|---|---|---|---|---|---|
| OML#6 | 7.1 | 6.0 | 5.9 | 6.4 | 5.8 | 5.8 | 28.1 | 6.2 | 628.8 |
| OML#156 | 4.5 | 4.2 | 3.7 | 4.4 | 4.0 | 4.2 | 17.5 | 4.2 | 60.2 |
| OML#44135 | 5.2 | 4.0 | 4.6 | 4.2 | 6.2 | 4.2 | 18.2 | 4.4 | 319.8 |
| MNIST | 17.6 | 10.4 | 11.3 | 11.3 | 10.4 | 12.1 | 49.6 | 12.5 | 27.9 |
| CIFAR10 | 106.0 | 99.6 | 101.0 | 106.1 | 97.5 | 108.0 | 496.0 | 102.2 | 6,436.5 |
| SVHN | 68.9 | 65.1 | 66.6 | 70.9 | 64.8 | 105.8 | 324.8 | 70.5 | 6,391.6 |
| CIFAR100 | 405.9 | 395.2 | 391.4 | 429.7 | 405.5 | 447.8 | 1,952.1 | 445.3 | - |
| T. ImageNet | 4,609.5 | 4,465.9 | 4,466.1 | 4,706.8 | 4,621.4 | 4,829.2 | 19,356.4 | 5,152.5 | - |
| FOOD101 | 4,464.8 | 4,339.8 | 4,350.3 | 4,475.7 | 4,471.0 | 4,726.8 | 18,903.3 | 4,703.9 | - |

**Runtime comparison *per* datatset** Table 2 presents the mean runtime (in minutes) for each algorithm on each dataset. Overall, compared to Entropy (considered one of the fastest), LDM-S incurs only a $3 \sim 6\%$ increase in runtime on CIFAR10, SVHN, CIFAR100, Tiny ImageNet, and FOOD101. LDM-S is slightly faster than MC-BALD, Coreset, and BADGE on some datasets. The more considerable runtime increase observed on OpenML and MNIST datasets is attributed to the relatively small training time compared to acquisition time due to simpler models and datasets. Due to individual training of all ensemble networks, ENS requires significantly more time (around 5x) than Entropy. BAIT exhibits runtime proportional to $d^2 C^2 q$, where $d, C$, and $q$ represent feature dimension, number of classes, and query size, respectively. This leads to significantly higher runtime, even for smaller datasets, making it impractical for large-scale tasks. In conclusion, LDM-S demonstrates superior or comparable performance to all considered baselines.

## 5 CONCLUSION

This paper introduces the least disagree metric (LDM), which measures uncertainty by considering the disagreement between the original and its perturbed model. Furthermore, the paper proposes a hypothesis sampling method for approximating the LDM and an LDM-based active learning algorithm (LDM-S). LDM-S selects unlabeled samples with small LDM values while also incorporating batch diversity. Extensive evaluations across various datasets demonstrate that LDM-S consistently achieves the best performance or remains comparable to other high-performing active learning algorithms. These results establish LDM-S as a state-of-the-art method in the field.

One immediate future direction is obtaining a rigorous sample complexity guarantee for LDM-S. This will provide a theoretical understanding of the number of samples required by LDM-S to achieve the desired level of accuracy. Exploring the integration of scalable and simple posterior sampling frameworks instead of the current Gaussian sampling scheme presents an exciting potential for improvement. This could lead to more efficient and practical implementations of LDM-S, particularly for larger datasets.

ACKNOWLEDGMENTS

We thank the anonymous reviewers for their helpful comments. We also thank Juho Lee and Chulhee Yun for their helpful discussions during the initial phase of this research. This work was partly supported by the Institute for Information & communications Technology Planning & Evaluation (IITP) grant funded by the Ministry of Science and ICT (MSIT) (No. 2021-0-01381 and 2022-0-00184) and partly supported by the Commercialization Promotion Agency for R&D Outcomes (COMPA) grant funded by MSIT (No. 1711198890). The research was conducted at the Korea Advanced Institute of Science and Technology (KAIST) and the Korea Institute of Oriental Medicine (KIOM) (No. NSN2221030).

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

## CONTENTS

APPENDIX

## A    RELATED WORK

There are various active learning algorithms such as uncertainty sampling (Lewis & Gale, 1994; Sharma & Bilgic, 2017), expected model change (Freytag et al., 2014; Ash et al., 2020), model output change (Mohamadi et al., 2022), expected error reduction (Yoo & Kweon, 2019; Zhao et al., 2021a), training dynamics (Wang et al., 2022), uncertainty reduction (Zhao et al., 2021b), core-set approach (Sener & Savarese, 2018; Mahmood et al., 2022; Yehuda et al., 2022), clustering (Yang et al., 2021; Citovsky et al., 2021), Bayesian active learning (Pinsler et al., 2019; Shi & Yu, 2019), discriminative sampling (Sinha et al., 2019; Zhang et al., 2020; Gu et al., 2021; Caramalau et al., 2021), constrained learning (Elenter et al., 2022), and data augmentation (Kim et al., 2021).

Various forms of uncertainty measures have been studied for the uncertainty-based approach. *Entropy* (Shannon, 1948) based algorithms query unlabeled samples yielding the maximum entropy from the predictive distribution. These algorithms perform poorly in multiclass as the entropy is heavily influenced by probabilities of less important classes (Joshi et al., 2009). *Margin* based algorithms (Balcan et al., 2007) query unlabeled samples closest to the decision boundary. The sample's closeness to the decision boundary is often not easily tractable in deep architecture (Mickisch et al., 2020). *Mutual information* based algorithms such as DBAL (Gal et al., 2017) and BatchBALD (Kirsch et al., 2019) query unlabeled samples yielding the maximum mutual information between predictions and posterior of model parameters. Both works use MC-dropout (Gal & Ghahramani, 2016) for deep networks to evaluate BALD (Houlsby et al., 2011). DBAL does not consider the correlation in the batch, and BatchBALD, which approximates batch-wise joint mutual information, is not appropriate for large query sizes. *Disagreement* based query-by-committee (QBC) algorithms (Beluch et al., 2018) query unlabeled samples yielding the maximum disagreement in labels predicted by the committee. It requires a high computational cost for individual training of each committee network. *Gradient* based algorithm BADGE (Ash et al., 2020) queries unlabeled samples that are likely to induce significant and diverse changes to the model, and *Fisher Information* (van der Vaart, 2000) based algorithm BAIT (Ash et al., 2021) queries unlabeled samples for which the resulting MAP estimate has the lowest Bayes risk using Fisher information matrix. Both BADGE and BAIT require a high computational cost when the feature dimension or the number of classes is large.

## B    PROOF OF THEOREM 1: LDM ESTIMATOR IS CONSISTENT

We consider multiclass classification, which we recall here from Section 2. Let $\mathcal{X}$ and $\mathcal{Y}$ be the instance and label space with $\mathcal{Y} = \{e_i\}_{i=1}^C$, where $e_i$ is the $i^{\text{th}}$ standard basis vector of $\mathbb{R}^C$ (i.e. one-hot encoding of the label $i$), and $\mathcal{H}$ be the hypothesis space of $h : \mathcal{X} \to \mathcal{Y}$.

By the triangle inequality, we have that

$$|L_{N,M} - L| \leq \underbrace{\left| L_{N,M} - \widetilde{L}_N \right|}_{\triangleq \Delta_1(N,M)} + \underbrace{\left| \widetilde{L}_N - L \right|}_{\triangleq \Delta_2(N)},$$

where we denote $\widetilde{L}_N := \min_{h \in \mathcal{H}_N^{g,x_0}} \rho(h,g)$.

We deal with $\Delta_1(N,M)$ first. By definition,

$$\Delta_1(N,M) = \left| \inf_{h \in \mathcal{H}_N^{g,x_0}} \rho_M(h,g) - \inf_{h \in \mathcal{H}_N^{g,x_0}} \rho(h,g) \right|$$

$$= \left| \inf_{h \in \mathcal{H}_N^{g,x_0}} \frac{1}{M} \sum_{i=1}^M \mathbb{I}\left[ h(X_i) \neq g(X_i) \right] - \inf_{h \in \mathcal{H}_N^{g,x_0}} \mathbb{E}_{X \sim D_{\mathcal{X}}}\left[ \mathbb{I}[h(X) \neq g(X)] \right] \right|$$

As $\Delta_1(N,M)$ is a difference of infimums of a sequence of functions, we need to establish a uniform convergence-type result over a sequence of sets. This is done by invoking "general" Glivenko-Cantelli (Lemma 1) and our generic bound (Lemma 2) on the empirical Rademacher complexity on

our hypothesis class $\mathcal{F} = \{f(\boldsymbol{x}) = \mathbb{I}[h(\boldsymbol{x}) \neq g(\boldsymbol{x})] \mid h \in \mathcal{H}_N^{g,\boldsymbol{x}_0}\}$, i.e., for any scalar $\delta \geq 0$, we have

$$\sup_{h \in \mathcal{H}_N^{g,\boldsymbol{x}_0}} |\rho_M(h,g) - \rho(h,g)| \leq 2\sqrt{\frac{2\log(CN)}{M}} + \delta. \tag{7}$$

with $\mathcal{D}_{\mathcal{X}}$-probability at least $1 - \exp\left(-\frac{M\delta^2}{8}\right)$.

Now let $\delta > 0$ be arbitrary, and for simplicity, denote $\delta' = 2\sqrt{\frac{2\log(CN)}{M}} + \delta$. As $\mathcal{H}_N^{g,\boldsymbol{x}_0}$ is finite, the infimums in the statement are achieved by some $g_1, g_2 \in \mathcal{H}_N^{g,\boldsymbol{x}_0}$, respectively. Then, due to the uniform convergence, we have that with probability at least $1 - e^{-\frac{M\delta^2}{8}}$,

$$\inf_{h \in \mathcal{H}_N^{g,\boldsymbol{x}_0}} \rho(h,g) = \rho(g_2,g) > \rho_M(g_2,g) - \delta' > \inf_{h \in \mathcal{H}_N^{g,\boldsymbol{x}_0}} \rho_M(h,g) - \delta'$$

and

$$\inf_{h \in \mathcal{H}_N^{g,\boldsymbol{x}_0}} \rho_M(h,g) = \rho_M(g_1,g) > \rho(g_1,g) - \delta' > \inf_{h \in \mathcal{H}_N^{g,\boldsymbol{x}_0}} \rho(h,g) - \delta',$$

and thus,

$$\Delta_1(N,M) = \left| \inf_{h \in \mathcal{H}_N^{g,\boldsymbol{x}_0}} \rho_M(h,g) - \inf_{h \in \mathcal{H}_N^{g,\boldsymbol{x}_0}} \rho(h,g) \right| \leq 2\sqrt{\frac{2\log(CN)}{M}} + \delta.$$

Choosing $M > \frac{8}{\delta^2}\log(CN)$, we have that $\Delta_1(N,M) \leq 2\delta$ with probability at least $1 - \frac{1}{CN}$.

We now deal with $\Delta_2(N)$. Recalling its definition, we have that for any $h^* \in \mathcal{H}^{g,\boldsymbol{x}_0}$ with $\rho(h^*,g) - L(g,\boldsymbol{x}_0) < \varepsilon$,

$$\Delta_2(N) = \left| \min_{h \in \mathcal{H}_N^{g,\boldsymbol{x}_0}} \rho(h,g) - \inf_{h \in \mathcal{H}^{g,\boldsymbol{x}_0}} \rho(h,g) \right|$$

$$\leq \left| \min_{h \in \mathcal{H}_N^{g,\boldsymbol{x}_0}} \rho(h,g) - \rho(h^*,g) \right| + |\rho(h^*,g) - L(g,\boldsymbol{x}_0)|$$

$$\leq \min_{h \in \mathcal{H}_N^{g,\boldsymbol{x}_0}} |\rho(h,g) - \rho(h^*,g)| + \varepsilon$$

$$\leq \min_{h \in \mathcal{H}_N^{g,\boldsymbol{x}_0}} \rho(h,h^*) + \varepsilon$$

$$\overset{(*)}{\leq} B \min_{h \in \mathcal{H}_N^{g,\boldsymbol{x}_0}} d_{\mathcal{H}}(h,h^*) + \varepsilon,$$

where $(*)$ follows from Assumption 2. Taking the infimum over all possible $h^*$ on both sides, by Assumption 3, we have that $\Delta_2(N) \leq \alpha(N,\varepsilon) + \varepsilon$ w.p. at least $1 - \beta(N)$.

Using the union bound, we have that with $M > \frac{8}{\delta^2}\log(CN)$,

$$\mathbb{P}\left[L_{N,M}(g,\boldsymbol{x}_0) - L(g,\boldsymbol{x}_0) \leq 2\delta + \alpha(N,\varepsilon) + \varepsilon\right] \geq \mathbb{P}\left[\Delta_1(N,M) + \Delta_2(N) \leq 2\delta + \alpha(N,\varepsilon) + \varepsilon\right]$$

$$\geq 1 - \frac{1}{CN} - \beta(N).$$

For the last part (convergence in probability), we start by denoting $Z_{\delta,N} = |L_{N,M}(g,\boldsymbol{x}_0) - L(g,\boldsymbol{x}_0)|$. (Recall that $M$ is a function of $\delta$, according to our particular choice). Under the prescribed limits and arbitrarities and our assumption that $\alpha(N), \beta(N) \to 0$ as $N \to \infty$, we have that for any sufficiently small $\delta' > 0$ sufficiently large $N$,

$$\mathbb{P}\left[Z_{\delta',s} \leq 2\delta'\right] \geq 1 - 2\delta'.$$

To be precise, given arbitrary $\theta_0, \Delta, \varepsilon > 0$, let $\delta' > 0$ be such that $2\delta + \varepsilon < \frac{\delta'}{2}$. Then it suffices to choose $N > \max\left(\alpha^{-1}\left(\frac{\delta'}{2};\varepsilon\right), \frac{2}{C\delta'} + \beta^{-1}\left(\frac{\delta'}{2}\right)\right)$, where $\alpha^{-1}(\cdot;\varepsilon)$ is the inverse function of $\alpha(\cdot,\varepsilon)$ w.r.t. the first argument, and $\beta^{-1}$ is the inverse function of $\beta$.

This implies that $d_{KF}(Z_{\delta',s}, 0) \le 2\delta'$, where $d_{KF}(X, Y) = \inf\{\delta' \ge 0 \mid \mathbb{P}[|X - Y| \ge \delta'] \le \delta'\}$ is the *Ky-Fan metric*, which induces a metric structure on the given probability space with the convergence in probability (see Section 9.2 of (Dudley, 2002)). As $\theta_0, \Delta, \varepsilon$ is arbitrary and thus so is $\delta'$, this implies that $|L_{N,M}(g, \boldsymbol{x}_0) - L(g, \boldsymbol{x}_0)| \overset{\mathbb{P}}{\to} 0$.

**Remark 2.** *We believe that Assumption 3 can be relaxed with a weaker assumption. This seems to be connected with the notion of $\epsilon$-net (Wainwright, 2019) as well as recent works on random Euclidean coverage (Reznikov & Saff, 2015; Krapivsky, 2023; Aldous, 2022; Penrose, 2023). Although the aforementioned works and our last assumption are common in that the random set is constructed from i.i.d. samples(hypotheses), one main difference is that for our purpose, instead of covering the entire space, one just needs to cover a certain portion at which the infimum is (approximately) attained.*

## B.1 SUPPORTING LEMMAS AND PROOFS

**Lemma 1** (Theorem 4.2 of Wainwright (2019)). *Let $Y_1, \cdots, Y_n \overset{i.i.d.}{\sim} \mathbb{P}$ for some distribution $\mathbb{P}$ over $\mathcal{X}$. For any $b$-uniformly bounded function class $\mathcal{F}$, any positive integer $n \ge 1$ and any scalar $\delta \ge 0$, we have*

$$\sup_{f \in \mathcal{F}} \left| \frac{1}{n} \sum_{i=1}^{n} f(Y_i) - \mathbb{E}[f(Y)] \right| \le 2\mathcal{R}_n(\mathcal{F}) + \delta$$

*with $\mathbb{P}$-probability at least $1 - \exp\left(-\frac{n\delta^2}{8b}\right)$. Here, $\mathcal{R}_n(\mathcal{F})$ is the empirical Rademacher complexity of $\mathcal{F}$.*

Recall that in our case, $n = M$, $b = 1$, and $\mathcal{F} = \{f(\boldsymbol{x}) = \mathbb{I}[h(\boldsymbol{x}) \ne g(\boldsymbol{x})] \mid h \in \mathcal{H}_N^{g;\boldsymbol{x}_0}\}$. The following lemma provides a generic bound on the empirical Rademacher complexity of $\mathcal{F}$:

**Lemma 2.** $\mathcal{R}_M(\mathcal{F}) \le \sqrt{\frac{2\log(CN)}{M}}$.

*Proof.* For simplicity, we denote $\mathbb{E} \triangleq \mathbb{E}_{\{X_i\}_{i=1}^M, \boldsymbol{\sigma}}$, where the expectation is w.r.t. $X_i \sim \mathcal{D}_{\mathcal{X}}$ i.i.d., and $\boldsymbol{\sigma}$ is the $M$-dimensional Rademacher variable. Also, let $l : [M] \to [C]$ be the labeling function for fixed samples $\{X_i\}_{i=1}^M$ i.e. $l(i) = \arg\max_{c \in [C]} [g(X_i)]_c$. As $g$ outputs one-hot encoding for each $i$, $l(i)$ is unique and thus well-defined. Thus, we have that $f(X_i) = \mathbb{I}[h(X_i) \ne g(X_i)] = 1 - h_{l(i)}(X_i)$, where we denote $h = (h_1, h_2, \cdots, h_C)$ with $h_j : \mathcal{X} \to \{0, 1\}$.

By definition,

$$
\begin{aligned}
\mathcal{R}_M(\mathcal{F}) &= \mathbb{E}\left[ \sup_{f \in \mathcal{F}} \frac{1}{M} \sum_{i=1}^{M} \sigma_i f(X_i) \right] \\
&= \mathbb{E}\left[ \sup_{h \in \mathcal{H}_N^{g,\boldsymbol{x}_0}} \frac{1}{M} \sum_{i=1}^{M} \sigma_i (1 - h_{l(i)}(X_i)) \right] \\
&= \mathbb{E}\left[ \sup_{h \in \mathcal{H}_N^{g,\boldsymbol{x}_0}} \frac{1}{M} \sum_{i=1}^{M} \sigma_i h_{l(i)}(X_i) \right] \\
&\le \mathbb{E}\left[ \sup_{h \in \mathcal{H}_N^{g,\boldsymbol{x}_0}, l \in [C]} \frac{1}{M} \sum_{i=1}^{M} \sigma_i h_l(X_i) \right] \\
&= \mathcal{R}_M\left( \{h_l \mid h \in \mathcal{H}_N^{g,\boldsymbol{x}_0}, l \in [C]\} \right) \\
&\le \sqrt{\frac{2\log(CN)}{M}},
\end{aligned}
$$

where the last inequality follows from Massart's Lemma (Massart, 2000) and the fact that $\mathcal{H}_N^{g;\boldsymbol{x}_0}$ is a finite set of cardinality at most $N$. $\qquad\square$

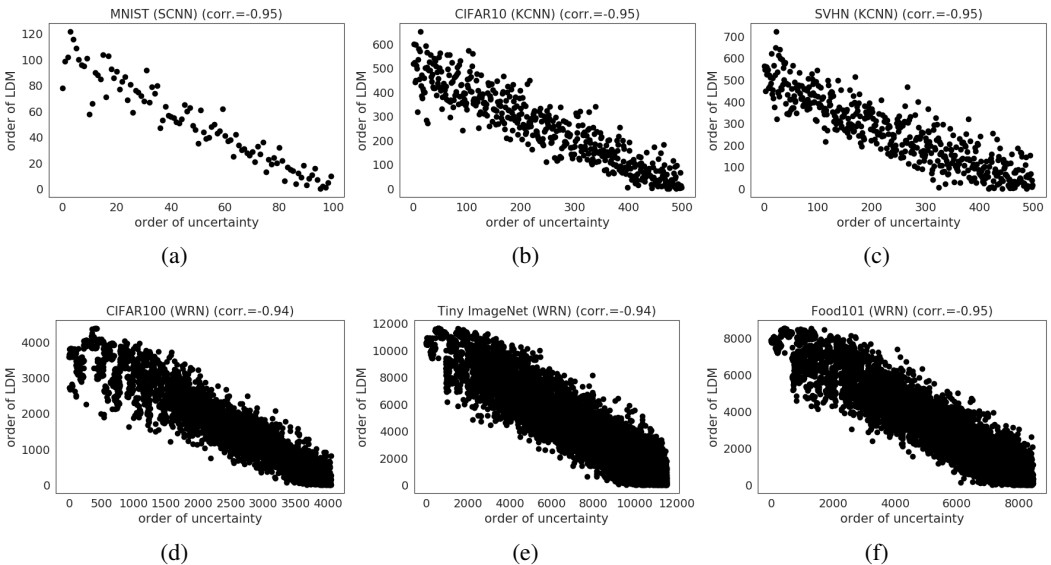

Figure 5: Examples of negative Spearman's rank correlation between LDM order and uncertainty order on MNIST (a), CIFAR10 (b), SVHN (c), CIFAR100 (d), Tiny ImageNet (e), and FOOD101 (f).

### B.2 PROOF OF COROLLARY 1

With the given assumptions, we know that $L_{M,N}(g, \boldsymbol{x}_k) \xrightarrow{\mathbb{P}} L(g, \boldsymbol{x}_k)$ for $k \in \{i, j\}$. For notation simplicity we denote $L(g, \boldsymbol{x}_k)$ as $L^{(k)}$, $L_{M,N}(g, \boldsymbol{x}_k)$ as $L_N^{(k)}$ and $\lim_{\substack{\min(M,N) \to \infty \\ M = \omega(\log(CN))}}$ as $\lim_{N \to \infty}$. For arbitrary $\varepsilon > 0$, we have

$$
\begin{aligned}
\lim_{N \to \infty} \mathbb{P}\left[L_N^{(i)} > L_N^{(j)} + \varepsilon\right] &= \lim_{N \to \infty} \mathbb{P}\left[L_N^{(i)} > L_N^{(j)} + \varepsilon \ \wedge \ |L_N^{(i)} - L^{(i)}| < \frac{\varepsilon}{2} \ \wedge \ |L_N^{(j)} - L^{(j)}| < \frac{\varepsilon}{2}\right] \\
&\leq \lim_{N \to \infty} \mathbb{P}\left[L^{(i)} + \frac{\varepsilon}{2} > L^{(j)} - \frac{\varepsilon}{2} + \varepsilon\right] \\
&= \mathbb{P}\left[L^{(i)} > L^{(j)}\right] = 0.
\end{aligned}
$$

## C THEORETICAL VERIFICATIONS OF INTUITIONS

Here, we consider two-dimensional binary classification with a set of linear classifiers, $\mathcal{H} = \{h : h(\boldsymbol{x}) = \text{sgn}(\boldsymbol{x}^\mathsf{T}\boldsymbol{w})\}$ where $\boldsymbol{w} \in \mathbb{R}^2$ is the parameter of $h$. We assume that $\boldsymbol{x}$ is uniformly distributed on $\mathcal{X} = \{\boldsymbol{x} : \|\boldsymbol{x}\| \leq 1\} \subset \mathbb{R}^2$. For a given $g \in \mathcal{H}$, let $\boldsymbol{v}$ be the parameter of $g$.

### C.1 LDM AS AN UNCERTAINTY MEASURE

Recall from Section 2.1 that the intuition behind a sample with a small LDM indicates that its prediction can be easily flip-flopped even by a small perturbation in the predictor. We theoretically prove this intuition

**Proposition 1.** *Suppose that $h$ is sampled with $\boldsymbol{w} \sim \mathcal{N}(\boldsymbol{v}, \mathbf{I}\sigma^2)$. Then,*

$$
L(g, \boldsymbol{x}_1) < L(g, \boldsymbol{x}_2) \Longleftrightarrow \mathbb{P}[h(\boldsymbol{x}_1) \neq g(\boldsymbol{x}_1)] > \mathbb{P}[h(\boldsymbol{x}_2) \neq g(\boldsymbol{x}_2)].
$$

Figure 5 shows examples of Spearman's rank correlation (Spearman, 1904) between the order of LDM and uncertainty, which is defined as the ratio of label predictions by a small perturbation in the predictor, on MNIST, CIFAR10, SVHN, CIFAR100, Tiny ImageNet, and FOOD101 datasets. Samples with increasing LDM or uncertainty are ranked from high to low. The results show that

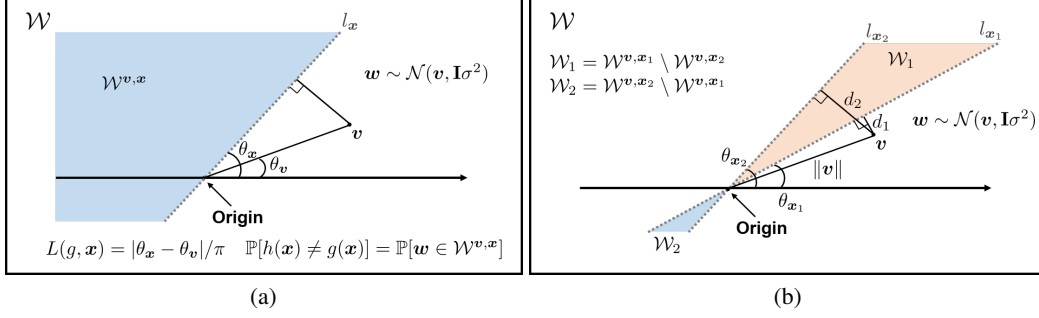

Figure 6: Proof of Proposition 1.

LDM and uncertainty orders have a strong negative rank correlation. Thus, a sample with a smaller LDM is closer to the decision boundary and is more uncertain.

*Proof of Proposition 1.* One important observation is that by the duality between $w$ and $x$ (Tong & Chang, 2001), in $\mathbb{R}^2$, $w$ is a point and $x$ is represented by the hyperplane, $l_x = \{w \in \mathbb{R}^2 : \text{sgn}(x^\top w) = 0\}$. Suppose that $h$ is sampled with $w \sim \mathcal{N}(v, \mathbf{I}\sigma^2)$, and let $\theta_v$ be the angle of $v$, $\theta_x$ be the angle between $l_x$ and positive x-axis, and $\mathcal{W}^{v,x}$ be the half-plane divided by $l_x$ which does not contain $v$:

$$\mathcal{W}^{v,x} = \{w' \in \mathcal{W} \mid h'(x) \neq g(x)\}$$

as in Figure 6a. Then, $L(g, x) = |\theta_x - \theta_v|/\pi$ and $\mathbb{P}[h(x) \neq g(x)] = \mathbb{P}[w \in \mathcal{W}^{v,x}]$.

Let $d_1, d_2$ be the distances between $v$ and $l_{x_1}, l_{x_2}$ respectively, and

$$\mathcal{W}_1 = \mathcal{W}^{v,x_1} \setminus \mathcal{W}^{v,x_2}, \quad \mathcal{W}_2 = \mathcal{W}^{v,x_2} \setminus \mathcal{W}^{v,x_1}$$

as in Figure 6b. Suppose that $d_1 < d_2$, then $|\theta_{x_1} - \theta_v| < |\theta_{x_2} - \theta_v|$ since $d_i = \|w\| \sin |\theta_{x_i} - \theta_v|$, and

$$\mathbb{P}[w \in \mathcal{W}^{v,x_1}] - \mathbb{P}[w \in \mathcal{W}^{v,x_2}] = \mathbb{P}[w \in \mathcal{W}_1] - \mathbb{P}[w \in \mathcal{W}_2] > 0$$

by the following:

$$\mathcal{W}^{v,x_1} = \mathcal{W}_1 \cup (\mathcal{W}^{v,x_1} \cap \mathcal{W}^{v,x_2}), \quad \mathcal{W}^{v,x_2} = \mathcal{W}_2 \cup (\mathcal{W}^{v,x_1} \cap \mathcal{W}^{v,x_2})$$

where $\mathcal{W}_1, \mathcal{W}_2$, and $\mathcal{W}^{v,x_1} \cap \mathcal{W}^{v,x_2}$ are disjoint. Note that $\mathcal{W}_1$ and $\mathcal{W}_2$ are one-to-one mapped by the symmetry at the origin. Still, the probabilities are different by the biased location of $v$, i.e., $\phi(w_1|v, \sigma^2) > \phi(w_2|v, \sigma^2)$ for all pairs of $(w_1, w_2) \in \mathcal{W}_1 \times \mathcal{W}_2$ that are symmetric at the origin. Here $\phi(\cdot|v, \sigma^2)$ is the probability density function of the bivariate normal distribution with mean $v$ and covariance $\sigma^2 \mathbf{I}$. Thus,

$$L(g, x_1) < L(g, x_2) \iff d_1 < d_2 \iff \mathbb{P}[h(x_1) \neq g(x_1)] > \mathbb{P}[h(x_2) \neq g(x_2)].$$

□

## C.2   Varying $\sigma^2$ in Algorithm 1

Recall from Section 2.2 that the intuition behind using multiple $\sigma^2$ is that it controls the trade-off between the probability of obtaining a hypothesis with a different prediction than that of $g$ and the scale of $\rho_S(h, g)$. Theoretically, we show the following for two-dimensional binary classification with linear classifiers:

**Proposition 2.** *Suppose that $h$ is sampled with $w \sim \mathcal{N}(v, \mathbf{I}\sigma^2)$ where $w, v \in \mathcal{W}$ are parameters of $h, g$ respectively, then $\mathbb{E}_w[\rho(h, g)]$ is continuous and strictly increasing with $\sigma$.*

Figure 7 shows the relationship between $\bar{\rho}_S(h, g)$ and $\log \sigma$ for MNIST, CIFAR10, SVNH, CIFAR100, Tiny ImageNet, and FOOD101 datasets where $h$ is sampled with $w \sim \mathcal{N}(v, \mathbf{I}\sigma^2)$ and $\bar{\rho}_S(h, g)$ is the mean of $\rho_S(h, g)$ for sampled $h$s. The $\bar{\rho}_S(h, g)$ is monotonically increasing with $\log \sigma$ in all experimental settings for general deep learning architectures.

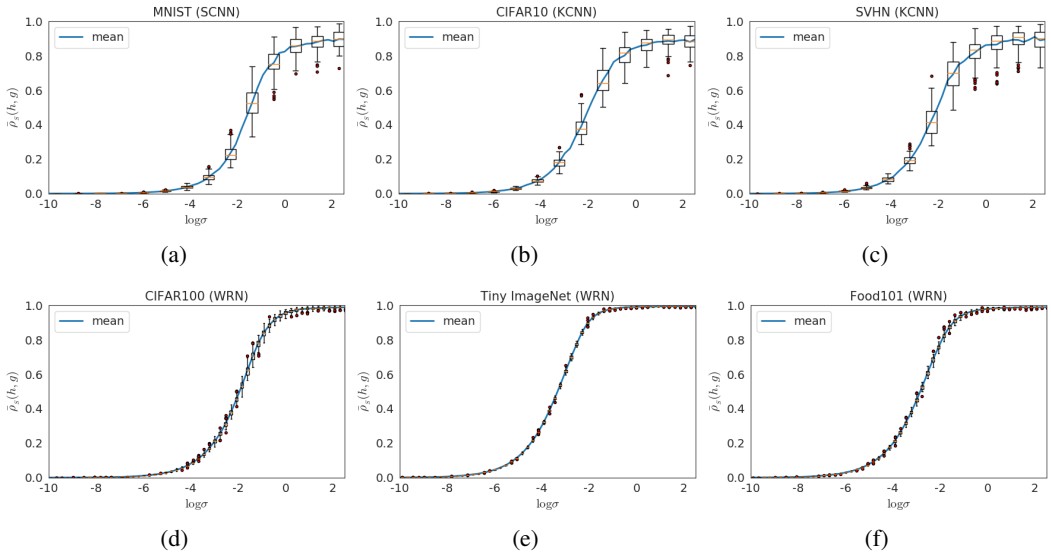

Figure 7: The relationship between the disagree metric and perturbation strength for MNIST (a), CIFAR10 (b), SVHN (c), CIFAR100 (d), Tiny ImageNet (e), and FOOD101 (f) datasets. $\bar{\rho}_S(h, g)$ is monotonically increasing with the perturbation strength in all experimental settings.

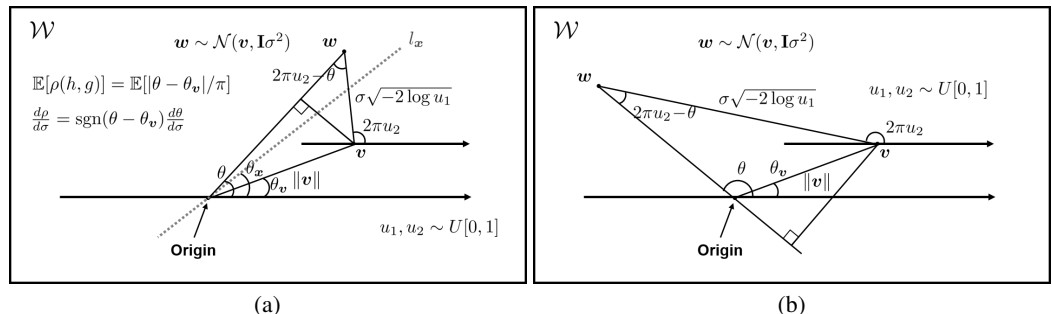

Figure 8: Proof of Proposition 2.

*Proof of Proposition 2.* By the duality between $\boldsymbol{w}$ and $\boldsymbol{x}$, in $\mathcal{W}$, $\boldsymbol{w}$ is a point and $\boldsymbol{x}$ is represented by the hyperplane, $l_{\boldsymbol{x}} = \{\boldsymbol{w} \in \mathcal{W} : \text{sgn}(\boldsymbol{x}^\mathsf{T}\boldsymbol{w}) = 0\}$. Let $h$ be a sampled hypothesis with $\boldsymbol{w} \sim \mathcal{N}(\boldsymbol{v}, \mathbf{I}\sigma^2)$, $\theta_{\boldsymbol{v}}$ be the angle of $\boldsymbol{v} = (v_1, v_2)^\mathsf{T}$, i.e., $\tan\theta_{\boldsymbol{v}} = v_2/v_1$, $\theta$ be the angle of $\boldsymbol{w} = (w_1, w_2)^\mathsf{T}$, i.e., $\tan\theta = w_2/w_1$, and $\theta_{\boldsymbol{x}}$ be the angle between $l_{\boldsymbol{x}}$ and positive x-axis. Here, $\theta, \theta_{\boldsymbol{x}} \in [-\pi + \theta_{\boldsymbol{v}}, \pi + \theta_{\boldsymbol{v}}]$ in convenience. When $\theta_{\boldsymbol{x}}$ or $\pi + \theta_{\boldsymbol{x}}$ is between $\theta$ and $\theta_{\boldsymbol{v}}$, $h(\boldsymbol{x}) \neq g(\boldsymbol{x})$, otherwise $h(\boldsymbol{x}) = g(\boldsymbol{x})$. Thus, $\rho(h, g) = |\theta - \theta_{\boldsymbol{v}}|/\pi$.

Using Box-Muller transform (Box & Muller, 1958), $\boldsymbol{w}$ can be generated by

$$w_1 = v_1 + \sigma\sqrt{-2\log u_1}\cos(2\pi u_2), \quad w_2 = v_2 + \sigma\sqrt{-2\log u_1}\sin(2\pi u_2)$$

where $u_1$ and $u_2$ are independent uniform random variables on $[0, 1]$. Then, $\|\boldsymbol{w} - \boldsymbol{v}\| = \sigma\sqrt{-2\log u_1}$ and $(w_2 - v_2)/(w_1 - v_1) = \tan(2\pi u_2)$, i.e., the angle of $\boldsymbol{w} - \boldsymbol{v}$ is $2\pi u_2$. Here,

$$\|\boldsymbol{v}\|\sin(\theta - \theta_{\boldsymbol{v}}) = \sigma\sqrt{-2\log u_1}\sin(2\pi u_2 - \theta) \tag{8}$$

by using the perpendicular line from $\boldsymbol{v}$ to the line passing through the origin and $\boldsymbol{w}$ (see the Figure 8 for its geometry), and Eq. 8 is satisfied for all $\theta$. For given $u_1$ and $u_2$, $\theta$ is continuous and the derivative of $\theta$ with respect to $\sigma$ is

$$\frac{d\theta}{d\sigma} = \frac{\sqrt{-2\log u_1}\sin^2(2\pi u_2 - \theta)}{\|\boldsymbol{v}\|\sin(2\pi u_2 - \theta_{\boldsymbol{v}})}, \quad \text{thus} \quad \begin{cases} \frac{d\theta}{d\sigma} > 0, & u_2 \in \left(\frac{\theta_{\boldsymbol{v}}}{2\pi}, \frac{\pi + \theta_{\boldsymbol{v}}}{2\pi}\right) \\ \frac{d\theta}{d\sigma} < 0, & u_2 \in [0, 1] \setminus \left[\frac{\theta_{\boldsymbol{v}}}{2\pi}, \frac{\pi + \theta_{\boldsymbol{v}}}{2\pi}\right] \end{cases}.$$

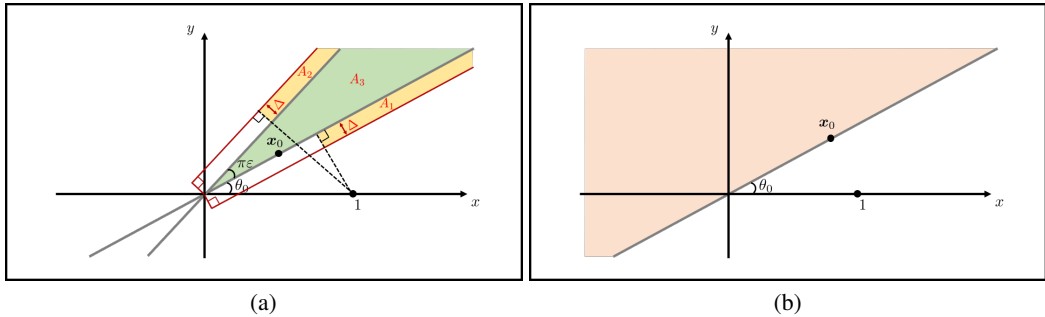

Figure 9: Proof of Proposition 3.

Then,

$$\frac{d\rho(h,g)}{d\sigma} = \text{sgn}(\theta - \theta_{\boldsymbol{v}})\frac{d\theta}{d\sigma} > 0 \quad \text{where} \quad u_2 \notin \left\{\frac{\theta_{\boldsymbol{v}}}{2\pi}, \frac{\pi + \theta_{\boldsymbol{v}}}{2\pi}\right\}.$$

Thus, $\rho(h,g)$ is continuous and strictly increasing with $\sigma$ when $u_2 \neq \frac{\theta_{\boldsymbol{v}}}{2\pi}$ and $u_2 \neq \frac{\pi + \theta_{\boldsymbol{v}}}{2\pi}$. Let $\rho(h,g) = F(\sigma, u_1, u_2)$, then $\mathbb{E}_{\boldsymbol{w}}[\rho(h,g)] = \int F(\sigma, u_1, u_2)f(u_1)f(u_2)du_1du_2$ where $f(u_i) = \mathbb{I}[0 < u_i < 1]$. For $0 < \sigma_1 < \sigma_2$,

$$\mathbb{E}_{\boldsymbol{w}\sim\mathcal{N}(\boldsymbol{v},\mathbf{I}\sigma_2^2)}[\rho(h,g)] - \mathbb{E}_{\boldsymbol{w}\sim\mathcal{N}(\boldsymbol{v},\mathbf{I}\sigma_1^2)}[\rho(h,g)]$$

$$= \int \left(F(\sigma_2, u_1, u_2) - F(\sigma_1, u_1, u_2)\right)f(u_1)f(u_2)du_1du_2 > 0.$$

$\square$

## C.3 ASSUMPTION 3 HOLDS WITH GAUSSIAN SAMPLING

**Proposition 3.** *Assume that the given sample $\boldsymbol{x}_0$ forms an angle $\theta_0 \in (0, \pi/2)$ w.r.t. the x-axis, and consider $\varepsilon \in (0,1)$ such that $\theta_0 + \pi\varepsilon < \frac{\pi}{2}$. Then, for binary classification with linear hypotheses and $\mathcal{H}_N$ comprising of $N$ i.i.d. random samples from $\mathcal{N}((1,0), \sigma^2 \boldsymbol{I}_2)$ for some fixed $\sigma^2 > 0$, Assumption 3 holds with $\alpha(N, \varepsilon) = \mathcal{O}\left(\frac{1}{\sqrt{N}}\right)$ and $\beta(N) = \mathcal{O}\left(e^{-\sqrt{N}}\right)$.*

*Proof.* We want to show that there exists $\alpha(N, \cdot)$ and $\beta(N)$ with $\lim_{N\to\infty}\alpha(N, \cdot) = \lim_{N\to\infty}\beta(N) = 0$ such that

$$\mathbb{P}\left[\inf_{\boldsymbol{w}^*:\theta(\boldsymbol{w}^*)\in(\theta_0,\theta_0+\pi\varepsilon)} \min_{\boldsymbol{w}\in\mathcal{H}_N^{g;\boldsymbol{x}_0}}\|\boldsymbol{w}^* - \boldsymbol{w}\|_2 \geq \frac{1}{B}\alpha(N, \varepsilon)\right] \leq \beta(N),$$

where $\theta(\boldsymbol{w}^*)$ is the angle made between $\boldsymbol{w}^*$ and the x-axis. Note that $\mathcal{H}_N^{g;\boldsymbol{x}_0}$ is random as well. Thus, we make use of a peeling argument as follows: conditioned on the event that $|\mathcal{H}_N^{g;\boldsymbol{x}_0}| = S$ for $S \in [N]$, we have that for any $\Delta \in (0, \sin\theta_0)$,

$$\mathbb{P}\left[\inf_{\boldsymbol{w}^*:\theta(h_{\boldsymbol{w}^*})\in(\theta_0,\theta_0+\pi\varepsilon)} \min_{\boldsymbol{w}\in\mathcal{H}_N^{g;\boldsymbol{x}_0}}\|\boldsymbol{w}^* - \boldsymbol{w}\|_2 \geq \Delta \,\middle|\, |\mathcal{H}_N^{g;\boldsymbol{x}_0}| = S\right] = (1 - p(\theta_0, \Delta, \varepsilon))^S,$$

where $p(\theta_0, \Delta, \varepsilon)$ is the measure of the region enclosed by the red boundary in Figure 9a with respect to $\mathcal{N}((1,0), \sigma^2\boldsymbol{I}_2)$. Also, we have that for $S \in [N]$

$$\mathbb{P}\left[|\mathcal{H}_N^{g;\boldsymbol{x}_0}| = S\right] = \binom{N}{S}q(\theta_0)^S(1 - q(\theta_0))^{N-S},$$

where $q(\theta_0)$ is the measure of the (light) red region in Figure 9b with respect to $\mathcal{N}((1,0), \sigma^2\boldsymbol{I}_2)$.

Thus, we have that

$$\mathbb{P}\left[\inf_{\boldsymbol{w}^*:\theta(\boldsymbol{w}^*)\in(\theta_0,\theta_0+\pi\varepsilon)} \min_{\boldsymbol{w}\in\mathcal{H}_N^{g;\boldsymbol{x}_0}}\|\boldsymbol{w}^* - \boldsymbol{w}\|_2 \geq \Delta\right]$$

$$= \sum_{S=0}^{N} \mathbb{P}\left[\inf_{\boldsymbol{w}^*:\theta(\boldsymbol{w}^*)\in(\theta_0,\theta_0+\pi\varepsilon)} \min_{\boldsymbol{w}\in\mathcal{H}_N^{g,\boldsymbol{x}_0}} \|\boldsymbol{w}^*-\boldsymbol{w}\|_2 \geq \Delta \,\middle|\, |\mathcal{H}_N^{g,\boldsymbol{x}_0}| = S\right] \mathbb{P}\left[|\mathcal{H}_N^{g,\boldsymbol{x}_0}| = S\right]$$

$$= \sum_{S=0}^{N} \binom{N}{S} q(\theta_0)^S (1-q(\theta_0))^{N-S} (1-p(\theta_0,\Delta,\varepsilon))^S$$

$$= ((1-q(\theta_0)) + q(\theta_0)(1-p(\theta_0,\Delta,\varepsilon)))^N$$

$$= (1 - p(\theta_0,\Delta,\varepsilon)q(\theta_0))^N$$

$$\leq e^{-Np(\theta_0,\Delta,\varepsilon)q(\theta_0)},$$

where the last inequality follows from the simple fact that $1 + x \leq e^x$ for any $x \in \mathbb{R}$.

Now it suffices to obtain non-vacuous lower bounds of $p(\theta_0,\Delta,\varepsilon)$ and $q(\theta_0)$.

**Lower bounding $q(\theta_0)$.** By rotational symmetry of $\mathcal{N}((1,0),\sigma^2\boldsymbol{I}_2)$ and the fact that rotation preserves Euclidean geometry, this is equivalent to finding the probability measure of the lower half-plane under the Gaussian distribution $\mathcal{N}((0,\sin\theta_0),\sigma^2\boldsymbol{I}_2)$, which is as follows:

$$q(\theta_0) = \frac{1}{2\pi\sigma^2}\int_{-\infty}^{0}\int_{-\infty}^{\infty} \exp\left(-\frac{x^2+(y-\sin\theta_0)^2}{2\sigma^2}\right) dxdy$$

$$= \frac{1}{\sqrt{2\pi}\sigma}\int_{-\infty}^{-\sin\theta_0} \exp\left(-\frac{y^2}{2\sigma^2}\right) dy$$

$$= \frac{1}{2} - \frac{1}{\sqrt{2\pi}\sigma}\int_{0}^{\sin\theta_0} \exp\left(-\frac{y^2}{2\sigma^2}\right) dy$$

$$\overset{(*)}{\geq} \frac{1}{2} - \frac{1}{\sqrt{2\pi}\sigma}\int_{0}^{\sin\theta_0} \left(\frac{2\sigma^2}{(\sin\theta_0)^2}\left(1-\exp\left(-\frac{(\sin\theta_0)^2}{2\sigma^2}\right)\right)y+1\right) dy$$

$$= \frac{1}{2} - \frac{1}{\sqrt{2\pi}\sigma}\left(\sigma^2\left(1-\exp\left(-\frac{(\sin\theta_0)^2}{2\sigma^2}\right)\right)+\sin\theta_0\right)$$

$$= \frac{1}{2} - \frac{1}{\sqrt{2\pi}}\left(\sigma\left(1-\exp\left(-\frac{(\sin\theta_0)^2}{2\sigma^2}\right)\right)+\frac{\sin\theta_0}{\sigma}\right),$$

where $(*)$ follows from the simple observation that $e^x \leq \frac{1-e^{-a}}{a}x + 1$ for $x \in [-a,0]$ and $a > 0$.

**Lower bounding $p(\theta_0,\Delta,\varepsilon)$.** Via similar rotational symmetry arguments and geometric decomposition of the region enclosed by the red boundary (see Figure 9a), we have that

$$p(\theta_0,\Delta,\varepsilon)$$
$$\geq A_1 + A_2 + A_3 \qquad\qquad\qquad\qquad \text{(see Figure 9a)}$$

$$\geq \frac{1}{2\pi\sigma^2}\int_{0}^{\Delta}\int_{0}^{\infty} \exp\left(-\frac{x^2+(y-\sin\theta_0)^2}{2\sigma^2}\right) dxdy$$

$$+ \frac{1}{2\pi\sigma^2}\int_{-\Delta}^{0}\int_{0}^{\infty} \exp\left(-\frac{x^2+(y-\sin(\theta_0+\pi\varepsilon))^2}{2\sigma^2}\right) dxdy$$

$$+ \frac{1}{2\pi\sigma^2}\int_{\theta_0}^{\theta_0+\pi\varepsilon}\int_{0}^{\infty} \exp\left(-\frac{(r\cos\theta-1)^2+(r\sin\theta)^2}{2\sigma^2}\right) rdrd\theta$$

$$= \frac{1}{2\sqrt{2\pi}\sigma}\int_{-\sin\theta_0}^{\Delta-\sin\theta_0} \exp\left(-\frac{y^2}{2\sigma^2}\right) dy + \frac{1}{2\sqrt{2\pi}\sigma}\int_{\sin(\theta_0+\pi\varepsilon)}^{\Delta+\sin(\theta_0+\pi\varepsilon)} \exp\left(-\frac{y^2}{2\sigma^2}\right) dy$$

$$+ \frac{1}{2\pi\sigma^2}\int_{\theta_0}^{\theta_0+\pi\varepsilon}\int_{0}^{\infty} \exp\left(-\frac{r^2-2r\cos\theta+1}{2\sigma^2}\right) rdrd\theta$$

$$\geq \frac{\Delta}{2\sqrt{2\pi}\sigma}\exp\left(-\frac{(\sin\theta_0)^2}{2\sigma^2}\right) + \frac{\Delta}{2\sqrt{2\pi}\sigma}\exp\left(-\frac{(\Delta+\sin(\theta_0+\pi\varepsilon))^2}{2\sigma^2}\right)$$

$$+ \frac{\varepsilon}{2\sigma^2}\int_{0}^{\infty} r\exp\left(-\frac{r^2+1}{2\sigma^2}\right) dr$$

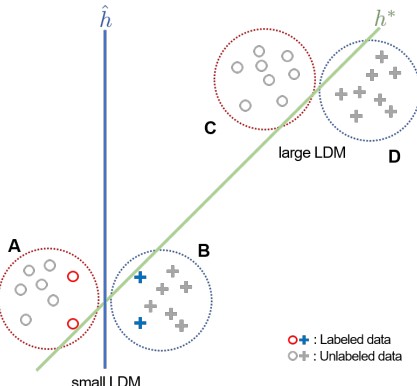

Figure 10: The effect of LDM-based seeding. Considering only LDM, samples are selected only in A or B, but samples in C or D are also selected if the seeding method is applied.

$$\geq \frac{\Delta}{\sqrt{2\pi}\sigma} \exp\left(-\frac{(\sin\theta_0 + 1)^2}{2\sigma^2}\right) + \frac{\varepsilon}{2} \exp\left(-\frac{1}{2\sigma^2}\right).$$

By choosing $\Delta = \mathcal{O}\left(\frac{1}{\sqrt{N}}\right)$, the proposition holds. $\qquad\square$

### C.4 Effect of Diversity in LDM-S

Here, we provide more intuition on why samples selected in the order of the smallest empirical LDM may not be the best strategy and, thus, why we need to pursue diversity via seeding, which was verified in realistic scenarios in Section 4.2. Again, let us consider a realizable binary classification with a set of linear classifiers, i.e., there exists $h^*$ whose test error is zero (See Figure 10). Let the red circles and blue crosses be the labeled samples and $g$ be the given hypotheses learned by the labeled samples. In this case, the LDMs of samples in groups A or B are small, and those in groups C or D are large. Thus, if we do not impose diversity and choose only samples with the smallest LDM, the algorithm will always choose samples in groups A or B. However, the samples in groups C or D are more helpful to us in this case, i.e., they provide more information. Therefore, pursuing diversity is necessary to provide the chance to query samples in C and D.

## D Datasets, Networks and Experimental Settings

### D.1 Datasets

**OpenML#6** (Frey & Slate, 1991) is a letter image recognition dataset which has $20,000$ samples in 26 classes. Each sample has 16 numeric features. In experiments, it is split into two parts: $16,000$ samples for training and $4,000$ samples for test.

**OpenML#156** (Vanschoren et al., 2014) is a synthesized dataset for random RBF which has $100,000$ samples in 5 classes. Each sample has 10 numeric features. In experiments, a subset is split into two parts: $40,000$ samples for training and $10,000$ samples for test.

**OpenML#44135** (Fanty & Cole, 1990) is a isolated letter speech recognition dataset which has $7,797$ samples in 26 classes. Each sample has 614 numeric features. In experiments, it is split into two parts: $6,237$ samples for training and $1,560$ samples for test.

**MNIST** (Lecun et al., 1998) is a handwritten digit dataset which has $60,000$ training samples and $10,000$ test samples in 10 classes. Each sample is a black and white image and $28 \times 28$ in size.

**CIFAR10** and **CIFAR100** (Krizhevsky, 2009) are tiny image datasets which has $50,000$ training samples and $10,000$ test samples in 10 and 100 classes respectively. Each sample is a color image and $32 \times 32$ in size.

Table 3: Settings for data and acquisition size. Acquisition size denotes the number of initial labeled samples + query size for each step (the size of pool data) → the number of final labeled samples.

| Dataset | Model | # of parameters sampled / total | Data size train / validation / test | Acquisition size | | |
|---|---|---|---|---|---|---|
| OpenML#6 | MLP | 3.4K/22.0K | 16,000 / - / 4,000 | 200 | +200 (2K) | → 4,000 |
| OpenML#156 | MLP | 0.6K/18.6K | 40,000 / - / 10,000 | 100 | +100 (2K) | → 2,000 |
| OpenML#44135 | MLP | 3.4K/98.5K | 6,237 / - / 1,560 | 100 | +100 (2K) | → 2,000 |
| MNIST | S-CNN | 1.3K/1.2M | 55,000 / 5,000 / 10,000 | 20 | +20 (2,000) | → 1,020 |
| CIFAR10 | K-CNN | 5.1K/2.2M | 45,000 / 5,000 / 10,000 | 200 | +400 (4,000) | → 9,800 |
| SVHN | K-CNN | 5.1K/2.2M | 68,257 / 5,000 / 26,032 | 200 | +400 (4,000) | → 9,800 |
| CIFAR100 | WRN-16-8 | 51.3K/11.0M | 45,000 / 5,000 / 10,000 | 5,000 | +2,000 (10,000) | → 25,000 |
| Tiny ImageNet | WRN-16-8 | 409.8K/11.4M | 90,000 / 10,000 / 10,000 | 10,000 | +5,000 (20,000) | → 50,000 |
| FOOD101 | WRN-16-8 | 206.9K/11.2M | 60,600 / 15,150 / 25,250 | 6,000 | +3,000 (15,000) | → 30,000 |
| ImageNet | ResNet-18 | 513K/11.7M | 1,153,047 / 128,120 / 50,000 | 128,120 | +64,060 (256,240) | → 384,360 |

Table 4: Settings for training.

| Dataset | Model | Epochs | Batch size | Optimizer | Learning Rate | Learning Rate Schedule ×decay [epoch schedule] |
|---|---|---|---|---|---|---|
| OpenML#6 | MLP | 100 | 64 | Adam | 0.001 | - |
| OpenML#156 | MLP | 100 | 64 | Adam | 0.001 | - |
| OpenML#44135 | MLP | 100 | 64 | Adam | 0.001 | - |
| MNIST | S-CNN | 50 | 32 | Adam | 0.001 | - |
| CIFAR10 | K-CNN | 150 | 64 | RMSProp | 0.0001 | - |
| SVHN | K-CNN | 150 | 64 | RMSProp | 0.0001 | - |
| CIFAR100 | WRN-16-8 | 100 | 128 | Nesterov | 0.05 | ×0.2 [60, 80] |
| Tiny ImageNet | WRN-16-8 | 200 | 128 | Nesterov | 0.1 | ×0.2 [60, 120, 160] |
| FOOD101 | WRN-16-8 | 200 | 128 | Nesterov | 0.1 | ×0.2 [60, 120, 160] |
| ImageNet | ResNet-18 | 100 | 128 | Nesterov | 0.001 | ×0.2 [60, 80] |

**SVHN** (Netzer et al., 2011) is a real-world digit dataset which has $73,257$ training samples and $26,032$ test samples in 10 classes. Each sample is a color image and $32 \times 32$ in size.

**Tiny ImageNet** (Le & Yang, 2015) is a subset of the ILSVRC (Russakovsky et al., 2015) dataset which has $100,000$ samples in 200 classes. Each sample is a color image and $64 \times 64$ in size. In experiments, Tiny ImageNet is split into two parts: $90,000$ samples for training and $10,000$ samples for test.

**FOOD101** (Bossard et al., 2014) is a fine-grained food image dataset which has $75,750$ training samples and $25,250$ test samples in 101 classes. Each sample is a color image resized to $75 \times 75$.

**ImageNet** (Russakovsky et al., 2015) is an image dataset organized according to the WordNet hierarchy, which has $1,281,167$ training samples and $50,000$ validation samples (we use the validation samples as test samples) in $1,000$ classes.

## D.2 DEEP NETWORKS

**MLP** consists of [128 dense - dropout (0.3) - 128 dense - dropout (0.3) - # class dense - softmax] layers, and it is used for OpenML datasets.

**S-CNN** (Chollet et al., 2015) consists of [$3 \times 3 \times 32$ conv $- 3 \times 3 \times 64$ conv $- 2 \times 2$ maxpool $-$ dropout $(0.25) - 128$ dense $-$ dropout $(0.5) - $ # class dense $-$ softmax] layers, and it is used for MNIST.

**K-CNN** (Chollet et al., 2015) consists of [two $3 \times 3 \times 32$ conv $- 2 \times 2$ maxpool - dropout $(0.25) -$ two $3 \times 3 \times 64$ conv $- 2 \times 2$ maxpool - dropout $(0.25) - 512$ dense $-$ dropout $(0.5) - $ # class dense - softmax] layers, and it is used for CIFAR10 and SVHN.

**WRN-16-8** (Zagoruyko & Komodakis, 2016) is a wide residual network that has 16 convolutional layers and a widening factor 8, and it is used for CIFAR100, Tiny ImageNet, and FOOD101.

**ResNet-18** (He et al., 2016) is a residual network that is a 72-layer architecture with 18 deep layers, and it is used for ImageNet.

### D.3 EXPERIMENTAL SETTINGS

The experimental settings for active learning regarding dataset, architecture, number of parameters, data size, and acquisition size are summarized in Table 3. Training settings regarding a number of epochs, batch size, optimizer, learning rate, and learning rate schedule are summarized in Table 4. The model parameters are initialized with He normal initialization (He et al., 2015) for all experimental settings. For all experiments, the initial labeled samples for each repetition are randomly sampled according to the distribution of the training set.

## E    PERFORMANCE PROFILE AND PENALTY MATRIX

### E.1    PERFORMANCE PROFILE

The performance profile, known as the Dolan-Moré plot, has been widely considered in benchmarking active learning (Tsymbalov et al., 2018; 2019), optimization profiles (Dolan & Moré, 2002), and even general deep learning tasks (Burnaev et al., 2015a;b). To introduce the Dolan-Moré plot, let $\text{acc}_A^{D,r,t}$ be the test accuracy of algorithm $A$ at step $t \in [T_D]$, for dataset $D$ and repetition $r \in [R]$, and $\Delta_A^{D,r,t} = \max_{A'}(\text{acc}_{A'}^{D,r,t}) - \text{acc}_A^{D,r,t}$. Here, $T_D$ is the number of steps for dataset $D$, and $R$ is the total number of repetitions. Then, we define the performance profile as

$$R_A(\delta) := \frac{1}{n_D} \sum_D \left[ \frac{\sum_{r,t} \mathbb{I}(\Delta_A^{D,r,t} \leq \delta)}{RT_D} \right],$$

where $n_D$ is the number of datasets. Intuitively, $R_A(\delta)$ is the fraction of cases where the performance gap between algorithm $A$ and the best competitor is less than $\delta$. Specifically, when $\delta = 0$, $R_A(0)$ is the fraction of cases on which algorithm $A$ performs the best.

### E.2    PENALTY MATRIX

The penalty matrix $P = (P_{ij})$ is evaluated as done in Ash et al. (2020): For each dataset, step, and each pair of algorithms $(A_i, A_j)$, we have 5 test accuracies $\{\text{acc}_i^r\}_{r=1}^5$ and $\{\text{acc}_j^r\}_{r=1}^5$ respectively. We compute the $t$-score as $t = \sqrt{5}\bar{\mu}/\bar{\sigma}$, where $\bar{\mu} = \frac{1}{5}\sum_{r=1}^5(\text{acc}_i^r - \text{acc}_j^r)$ and $\bar{\sigma} = \sqrt{\frac{1}{4}\sum_{r=1}^5(\text{acc}_i^r - \text{acc}_j^r - \bar{\mu})^2}$. The two-sided paired sample $t$-test is performed for the null that there is no performance difference between algorithms: $A_i$ is said to *beat* $A_j$ when $t > 2.776$ (the critical point of $p$-value being 0.05), and vice-versa when $t < -2.776$. Then, when $A_i$ beats $A_j$, we accumulate a penalty[4] of $1/T_D$ to $P_{i,j}$ where $T_D$ is the number of steps for dataset $D$, and vice-versa. Summing across the datasets gives us the final penalty matrix.

## F    ABLATION STUDY

### F.1    CHOICE OF STOP CONDITION $s$

Figure 11a shows the empirically evaluated LDM by Algorithm 1 in binary classification with the linear classifier as described in Figure 1. In the experiment, the true LDM of the sample is set to 0.01. The evaluated LDM is close to the true LDM when $s = 10$ and reaches the true LDM when $s \geq 20$ with a gap of roughly $10^{-4}$. This suggests that even with a moderate $s$, Algorithm 1 can approximate the true LDM with sufficiently low error.

Figure 11b shows the empirically evaluated LDMs of MNIST samples for a four-layered CNN where $M$ is set to be the total number of samples in MNIST, which is 60000. We denote $x_i$ as the $i^{\text{th}}$ sample *ordered* by the final evaluated LDM. Observe that the evaluated LDMs are monotonically decreasing as $s$ increases, and they seem to converge while maintaining the rank order. In practice, obtaining values close to the true LDM requires a large $s$, which is computationally prohibitive as the algorithm requires a considerable runtime to sample many hypotheses. For example, when $s = 50000$, our algorithm samples ∼50M hypotheses and takes roughly 18 hours to run and evaluate LDM. Therefore,

---

[4]This choice of penalty ensures that each dataset contributes equally to the resulting penalty matrix.

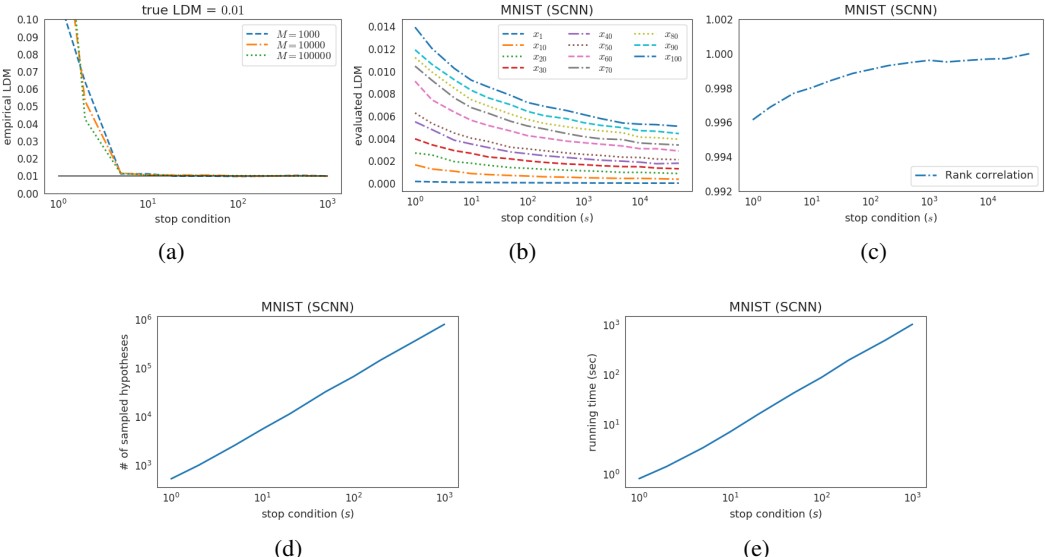

Figure 11: Empirically evaluated LDMs by Algorithm 1 with respect to the stop condition $s$. (a) Here, we consider the two-dimensional binary classification with the linear classifier (see Figure 1). The evaluated LDM is close to the true LDM even when $s = 10$ and reaches the true LDM when $s \geq 20$. (b) Evaluated LDMs of MNIST samples with a four-layered CNN. Observe that the evaluated LDM monotonically decreases as $s$ increases, and the rank order is well maintained. (c) In the same setting, the rank correlation coefficient of the evaluated LDMs at various $s$s to that at $s = 50000$. Note that already at $s = 10$, the rank correlation coefficient is 0.998, suggesting that $s = 10$ suffices. (d-e) For evaluating LDM, the number of sampled hypotheses and runtime are almost linearly proportional to the stop condition.

based upon the observation that the rank order is preserved throughout the values of $s$, we focus on the rank order of the evaluated LDMs rather than their actual values.

Figure 11c shows the rank correlation coefficient of the evaluated LDMs of a range of $s$'s to that at $s = 50000$. Even when $s = 10$, the rank correlation coefficient between $s = 10$ and $s = 50000$ is already 0.998. We observed that the evaluated LDMs' properties regarding the stop condition $s$ also hold for other datasets, i.e., the preservation of rank order holds in general.

Figure 11d and 11e show the number of sampled hypotheses and runtime with respect to the stop condition when LDM is evaluated. Both are almost linearly proportional to the stop condition and thus, we should set the stop condition to be as small as possible to reduce the running time in LDM evaluation.

### F.2 EFFECTIVENESS OF LDM

To isolate the effectiveness of LDM, we consider three other variants of LDM-S. 'LDM-smallest' select batches with the smallest LDMs *without* taking diversity into account, 'Seeding (cos)' and 'Seeding ($\ell_2$)' are the unweighted $k$-means++ seeding methods using cosine and $\ell_2$ distance, respectively. Note that the last two do not use LDM in any way. We have excluded batch diversity to clarify the effectiveness of LDM further.

Figure 12 shows the test accuracy with respect to the number of labeled samples on MNIST, CIFAR10, and CIFAR100 datasets. Indeed, we observe a significant performance improvement when using LDM and a further improvement when batch diversity is considered. Additional experiments are conducted to compare the $k$-means++ seeding with FASS (Wei et al., 2015) or Cluster-Margin (Citovsky et al., 2021), which can be a batch diversity method for LDM. Although FASS helps LDM slightly, it falls short of LDM-S. Cluster-Margin also does not help LDM and, surprisingly, degrades the performance. We believe this is because Cluster-Margin *strongly* pursues batch diversity, diminishing the effect of LDM as an uncertainty measure. Specifically, Cluster-Margin considers samples of varying LDM

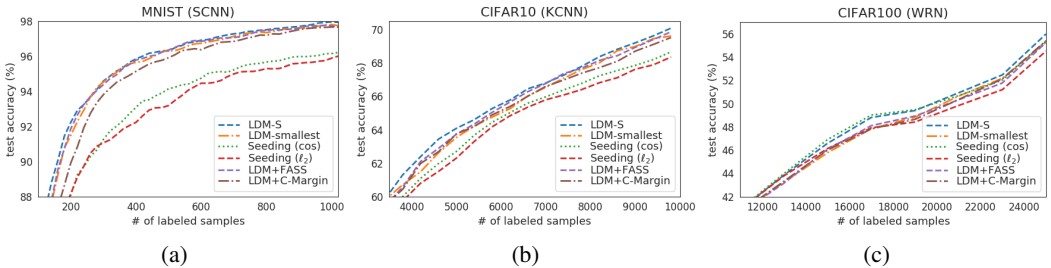

Figure 12: The effect of diverse sampling in LDM-S on MNIST (a), CIFAR10 (b), and CIFAR100 (c) datasets. 'LDM-smallest': selecting batch with the smallest LDM, 'Seeding (cos)': unweighted seeding using cosine distance, 'Seeding ($\ell_2$)': unweighted seeding using $\ell_2$-distance, 'LDM+FASS': the combination of LDM and FASS, 'LDM+C-Margin': the combination of LDM and Cluster-Margin. LDM-S leads to significant performance improvement compared to those without batch diversity, with FASS, or with Cluster-Margin.

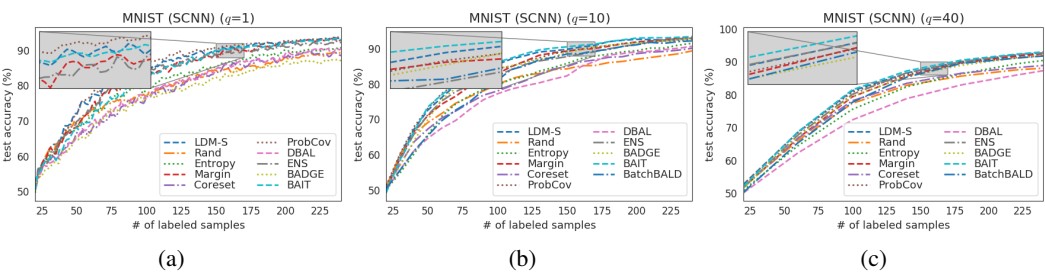

Figure 13: Performance comparison when the batch sizes are 1 (a), 10 (b), and 40 (c) on MNIST.

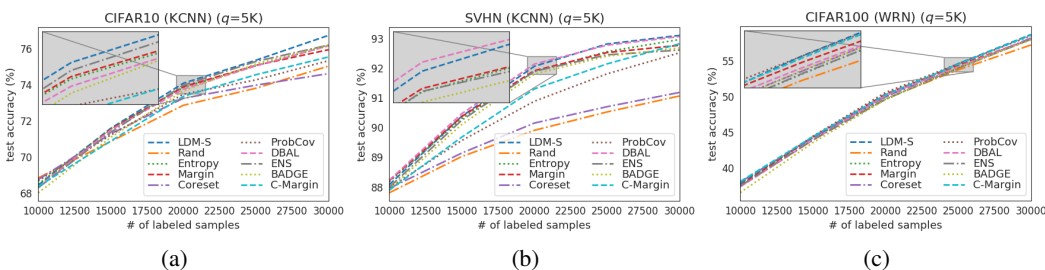

Figure 14: Performance comparison when the batch size is 5K on CIFAR10 (a), SVHN (b), and CIFAR100 (c).

scores from the beginning (a large portion of them are thus not so helpful). In contrast, our algorithm significantly weights the samples with small LDM, biasing our samples towards more uncertain samples (and thus more helpful).

## F.3   EFFECT OF BATCH SIZE

To verify the effectiveness of LDM-S for batch mode, we compare the active learning performance with respect to various batch sizes. Figure 13 shows the test accuracy when the batch sizes are 1, 10, and 40 on the MNIST dataset. Figure 14 shows the test accuracy when the batch size is 5K on CIFAR10, SVHM, and CIFAR100 datasets. Overall, LDM-S performs well compared to other algorithms, even with small and large batch sizes. Therefore, the proposed algorithm is robust to batch size, while other baseline algorithms often are not. For example, BADGE performs well with batch sizes 10 or 40 but is poor with batch size one on MNIST. Note that we have added additional

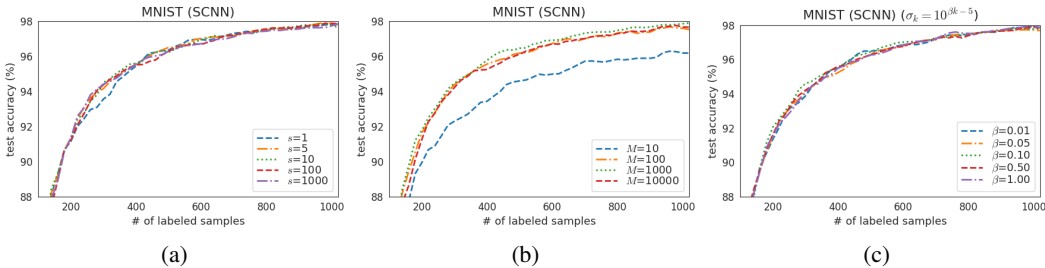

Figure 15: Performance vs hyperparameters. The test accuracy with respect to stop condition $s$ (a), the number of Monte Carlo samples for approximating $\rho$ (b), and sigmas' interval (c).

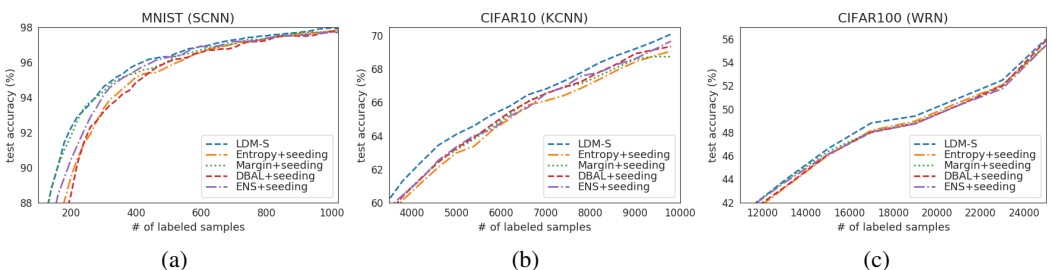

Figure 16: The performance comparison of LDM-S with the standard uncertainty methods to which weighted seeding is applied on MNIST (a), CIFAR10 (b), and CIFAR100 (c). Even if weighted seeding is applied to the standard uncertainty methods, LDM-S performs better.

results for BatchBALD in Figure 13 and Cluster-Margin (C-Margin) in Figure 14. For both settings, we've matched the setting as described in their original papers.

### F.4 EFFECT OF HYPERPARAMETERS IN LDM-S

There are three hyperparameters in the proposed algorithm: stop condition $s$, the number of Monte Carlo samples $M$, and the set of variances $\{\sigma_k^2\}_{k=1}^K$.

The stop condition $s$ is required for LDM evaluation. We set $s = 10$, considering the rank correlation coefficient and computing time. Figure 15a shows the test accuracy with respect to $s \in \{1, 5, 10, 100, 1000\}$, and there is no significant performance difference.

The number of Monte Carlo samples $M$ is set for approximating $\rho$. The proposed algorithm aims to distinguish LDMs of pool data. Thus, we set $M$ to be the same as the pool size. Figure 15b shows the test accuracy with respect to $M \in \{10, 100, 1000, 10000\}$, and there is no significant performance difference except where $M$ is extremely small, e.g., $M = 10$.

The set of variances $\{\sigma_k^2\}_{k=1}^K$ is set for hypothesis sampling. Figure 7 in Appendix C.2 shows the relationship between the disagree metric and $\sigma^2$. To properly approximate LDM, we need to sample hypotheses with a wide range of $\rho$, and thus, we need a wide range of $\sigma^2$. To efficiently cover a wide range of $\sigma^2$, we make the exponent equally spaced such as $\sigma_k = 10^{\beta k - 5}$ where $\beta > 0$ and set $\sigma$ the have $10^{-5}$ to 1. Figure 15c shows the test accuracy with respect to $\beta \in \{0.01, 0.05, 0.1, 0.5, 1\}$, and there is no significant performance difference.

## G ADDITIONAL RESULTS

### G.1 COMPARING WITH OTHER UNCERTAINTY METHODS WITH SEEDING

To clarify whether LDM-S's gains over the standard uncertainty methods are due to weighted seeding or to LDM's superiority, LDM-S's performance is compared with those methods to which weighted

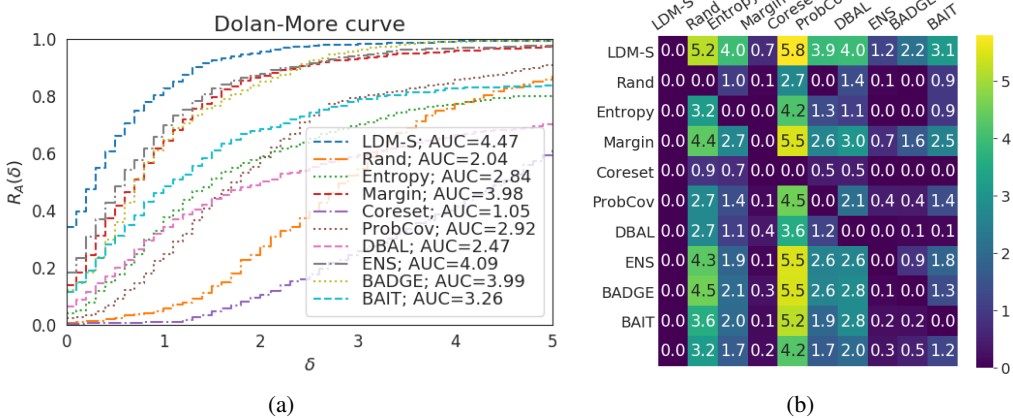

Figure 17: Comparison with BAIT across OpenML #6, #156, #44135, MNIST, CIFAR10, and SVHN datasets. (a) Dolan-Moré plot. (b) The penalty matrix.

seeding is applied. Figure 16 shows the test accuracy with respect to the number of labeled samples on MNIST, CIFAR10, and CIFAR100 datasets. Overall, even when weighted seeding is applied to the standard uncertainty methods, LDM-S still performs better on all datasets. Therefore, the performance gains of LDM-S can be attributed to LDM's superiority over the standard uncertainty measures.

## G.2 COMPARISON *across* DATASETS WITH BAIT

The performance profile and penalty matrix between LDM-S and BAIT are examined on OpenML #6, #156, #44135, MNIST, CIFAR10, and SVHN datasets where the experiments for BAIT are conducted. Figure 17a shows the performance profile w.r.t. $\delta$. LDM-S retains the highest $R_A(\delta)$ over all considered $\delta$'s, which means that our LDM-S outperforms the other algorithms, including BAIT. Figure 17b shows the penalty matrix. It is also clear that LDM-S generally outperforms all other algorithms, including BAIT.

## G.3 COMPARISON *per* DATASETS

Table 5 shows the mean $\pm$ std of the test accuracies (%) w.r.t. the number of labeled samples for OpenML #6, #156, #44135 with MLP; MNIST with SCNN; CIFAR10 and SVHN with K-CNN; CIFAR100, Tiny ImageNet, FOOD101 with WRN-16-8; and ImageNet with ResNet-18. Overall, LDM-S either consistently performs best or is at par with other algorithms for all datasets, while the performance of the algorithms except LDM-S varies depending on datasets.

Table 5: The mean ± std of the test accuracies (%) w.r.t. the number of labeled samples. (**bold+underlined**: best performance, **bold**: second-best performance)

| $|\mathcal{L}|$ | LDM-S | Rand | Entropy | Margin | Coreset | ProbCov | DBAL | ENS | BADGE | BAIT |
|---|---|---|---|---|---|---|---|---|---|---|
| **OpenML#6** | | | | | | | | | | |
| 400 | **_62.42 ± 0.01_** | 60.64 ± 0.01 | 56.92 ± 0.01 | 61.22 ± 0.03 | 61.23 ± 0.02 | 61.00 ± 0.01 | 58.64 ± 0.02 | 60.96 ± 0.02 | **61.65 ± 0.01** | 61.43 ± 0.01 |
| 800 | **_76.29 ± 0.01_** | 72.51 ± 0.00 | 66.38 ± 0.01 | 74.94 ± 0.01 | 73.66 ± 0.01 | 73.10 ± 0.00 | 70.59 ± 0.01 | **75.53 ± 0.01** | 75.13 ± 0.01 | 73.55 ± 0.00 |
| 1,200 | **_81.62 ± 0.01_** | 77.49 ± 0.00 | 70.62 ± 0.02 | 80.57 ± 0.00 | 78.64 ± 0.01 | 77.71 ± 0.01 | 75.98 ± 0.01 | **81.30 ± 0.01** | 79.85 ± 0.01 | 78.52 ± 0.01 |
| 1,600 | **_84.74 ± 0.00_** | 80.20 ± 0.00 | 73.61 ± 0.02 | 83.91 ± 0.01 | 81.26 ± 0.01 | 80.15 ± 0.01 | 79.07 ± 0.00 | **84.24 ± 0.01** | 82.60 ± 0.00 | 81.84 ± 0.00 |
| 2,000 | **_87.05 ± 0.00_** | 82.06 ± 0.00 | 77.49 ± 0.02 | 86.41 ± 0.00 | 83.17 ± 0.01 | 82.21 ± 0.01 | 81.73 ± 0.00 | **86.82 ± 0.00** | 84.81 ± 0.00 | 84.39 ± 0.01 |
| 2,400 | **_89.12 ± 0.00_** | 83.62 ± 0.01 | 81.80 ± 0.01 | 88.59 ± 0.01 | 85.00 ± 0.01 | 84.01 ± 0.01 | 84.20 ± 0.01 | **88.74 ± 0.01** | 87.04 ± 0.00 | 86.73 ± 0.01 |
| 2,800 | **_90.35 ± 0.00_** | 84.77 ± 0.01 | 85.12 ± 0.01 | 89.76 ± 0.00 | 86.22 ± 0.00 | 85.04 ± 0.00 | 86.00 ± 0.01 | **90.10 ± 0.01** | 88.31 ± 0.00 | 88.44 ± 0.00 |
| 3,200 | **_91.06 ± 0.00_** | 85.96 ± 0.01 | 87.37 ± 0.00 | 90.83 ± 0.00 | 87.22 ± 0.00 | 85.95 ± 0.00 | 87.43 ± 0.01 | **90.96 ± 0.00** | 89.18 ± 0.00 | 89.76 ± 0.00 |
| 3,600 | **_91.80 ± 0.00_** | 86.77 ± 0.00 | 88.57 ± 0.00 | 91.50 ± 0.00 | 87.85 ± 0.01 | 86.90 ± 0.01 | 88.83 ± 0.00 | **91.70 ± 0.00** | 90.21 ± 0.00 | 90.57 ± 0.00 |
| 4,000 | **_92.53 ± 0.00_** | 88.00 ± 0.01 | 89.54 ± 0.00 | 92.36 ± 0.00 | 88.49 ± 0.01 | 87.72 ± 0.00 | 89.69 ± 0.00 | **92.62 ± 0.00** | 91.35 ± 0.00 | 91.28 ± 0.00 |
| **OpenML#156** | | | | | | | | | | |
| 200 | 69.78 ± 0.02 | **70.30 ± 0.02** | 65.23 ± 0.02 | 67.95 ± 0.03 | 61.15 ± 0.03 | 70.02 ± 0.01 | 64.10 ± 0.04 | 67.97 ± 0.02 | **_71.35 ± 0.02_** | 67.62 ± 0.03 |
| 400 | 83.08 ± 0.02 | **83.27 ± 0.01** | 76.42 ± 0.03 | 81.58 ± 0.02 | 63.92 ± 0.03 | 82.68 ± 0.01 | 70.78 ± 0.01 | 82.19 ± 0.02 | **_83.74 ± 0.02_** | 76.04 ± 0.02 |
| 600 | **87.30 ± 0.01** | 86.20 ± 0.01 | 83.11 ± 0.01 | **86.75 ± 0.01** | 65.46 ± 0.04 | 85.65 ± 0.00 | 72.06 ± 0.02 | 86.38 ± 0.01 | 86.68 ± 0.01 | 77.68 ± 0.01 |
| 800 | **88.60 ± 0.00** | 87.09 ± 0.00 | 85.95 ± 0.01 | 88.14 ± 0.00 | 65.40 ± 0.05 | 86.79 ± 0.00 | 72.88 ± 0.02 | **88.17 ± 0.00** | 87.97 ± 0.00 | 78.47 ± 0.02 |
| 1,000 | **89.32 ± 0.00** | 87.63 ± 0.01 | 87.65 ± 0.01 | 88.83 ± 0.00 | 65.59 ± 0.05 | 87.56 ± 0.01 | 72.82 ± 0.02 | **89.21 ± 0.00** | 88.94 ± 0.00 | 78.89 ± 0.01 |
| 1,200 | **90.08 ± 0.00** | 88.31 ± 0.00 | 89.13 ± 0.00 | 89.83 ± 0.00 | 67.90 ± 0.05 | 88.51 ± 0.00 | 72.55 ± 0.03 | **89.88 ± 0.00** | 89.59 ± 0.00 | 79.29 ± 0.01 |
| 1,400 | **90.43 ± 0.00** | 88.78 ± 0.00 | 89.88 ± 0.00 | **90.40 ± 0.00** | 69.14 ± 0.02 | 89.03 ± 0.00 | 72.87 ± 0.02 | 90.23 ± 0.00 | 89.99 ± 0.00 | 80.45 ± 0.01 |
| 1,600 | **90.70 ± 0.00** | 89.11 ± 0.00 | 90.14 ± 0.00 | **90.67 ± 0.00** | 70.02 ± 0.04 | 89.46 ± 0.00 | 73.06 ± 0.01 | 90.48 ± 0.00 | 90.28 ± 0.00 | 80.70 ± 0.01 |
| 1,800 | **90.95 ± 0.00** | 89.42 ± 0.00 | 90.37 ± 0.00 | **90.85 ± 0.00** | 70.03 ± 0.05 | 89.69 ± 0.00 | 73.96 ± 0.02 | 90.75 ± 0.00 | 90.49 ± 0.00 | 81.85 ± 0.02 |
| 2,000 | **91.19 ± 0.00** | 89.67 ± 0.00 | 90.62 ± 0.00 | **91.09 ± 0.00** | 70.92 ± 0.05 | 90.05 ± 0.00 | 73.92 ± 0.03 | 91.01 ± 0.00 | 90.73 ± 0.00 | 84.17 ± 0.02 |
| **OpenML#44135** | | | | | | | | | | |
| 200 | **82.28 ± 0.01** | 77.44 ± 0.02 | 75.67 ± 0.01 | 81.33 ± 0.01 | 75.92 ± 0.00 | **_81.70 ± 0.01_** | 76.56 ± 0.02 | 78.16 ± 0.02 | 78.33 ± 0.01 | 78.82 ± 0.01 |
| 400 | **92.65 ± 0.01** | 86.21 ± 0.01 | 86.18 ± 0.02 | **92.10 ± 0.00** | 84.92 ± 0.01 | 91.59 ± 0.01 | 86.56 ± 0.01 | 88.60 ± 0.00 | 88.60 ± 0.00 | 89.50 ± 0.01 |
| 600 | **94.51 ± 0.00** | 89.18 ± 0.01 | 91.17 ± 0.01 | **94.12 ± 0.00** | 89.48 ± 0.01 | 93.51 ± 0.00 | 90.63 ± 0.00 | 92.18 ± 0.00 | 92.06 ± 0.01 | 92.29 ± 0.00 |
| 800 | **95.34 ± 0.01** | 91.05 ± 0.01 | 93.26 ± 0.00 | **95.30 ± 0.00** | 91.29 ± 0.01 | 94.69 ± 0.00 | 92.65 ± 0.00 | 93.70 ± 0.01 | 93.42 ± 0.00 | 93.49 ± 0.01 |
| 1,000 | **95.86 ± 0.00** | 91.90 ± 0.00 | 94.41 ± 0.00 | **96.03 ± 0.00** | 92.47 ± 0.00 | 95.02 ± 0.01 | 93.78 ± 0.00 | 94.82 ± 0.01 | 94.28 ± 0.01 | 94.41 ± 0.01 |
| 1,200 | **96.13 ± 0.00** | 92.35 ± 0.00 | 95.13 ± 0.00 | **96.28 ± 0.00** | 93.03 ± 0.00 | 95.22 ± 0.00 | 94.57 ± 0.01 | 95.29 ± 0.01 | 95.12 ± 0.00 | 94.87 ± 0.00 |
| 1,400 | **96.27 ± 0.00** | 92.81 ± 0.00 | 95.62 ± 0.00 | **96.38 ± 0.00** | 93.61 ± 0.00 | 95.57 ± 0.00 | 95.26 ± 0.00 | 95.53 ± 0.01 | 95.53 ± 0.00 | 95.43 ± 0.00 |
| 1,600 | **96.51 ± 0.00** | 93.29 ± 0.00 | 95.93 ± 0.00 | **96.44 ± 0.01** | 94.18 ± 0.01 | 95.73 ± 0.00 | 95.48 ± 0.00 | 95.58 ± 0.00 | 95.88 ± 0.01 | 95.64 ± 0.01 |
| 1,800 | **96.60 ± 0.00** | 93.42 ± 0.01 | 95.84 ± 0.01 | **96.38 ± 0.00** | 94.41 ± 0.00 | 95.79 ± 0.00 | 95.63 ± 0.00 | 95.72 ± 0.01 | 95.98 ± 0.00 | 95.84 ± 0.00 |
| 2,000 | **_96.62 ± 0.00_** | 93.73 ± 0.01 | **96.40 ± 0.00** | 96.21 ± 0.00 | 94.59 ± 0.00 | 95.96 ± 0.01 | 96.21 ± 0.00 | 96.13 ± 0.00 | 96.26 ± 0.00 | 95.65 ± 0.00 |

Continued on next page

Table 5 – Continued from previous page

| $|\mathcal{L}|$ | LDM-S | Rand | Entropy | Margin | Coreset | ProbCov | DBAL | ENS | BADGE | BAIT |
|---|---|---|---|---|---|---|---|---|---|---|
| **MNIST** | | | | | | | | | | |
| 120 | **87.08 ± 0.01** | 81.32 ± 0.02 | 80.67 ± 0.04 | 84.86 ± 0.01 | 79.29 ± 0.03 | 86.61 ± 0.01 | 78.59 ± 0.01 | 85.34 ± 0.01 | 85.43 ± 0.02 | **87.07 ± 0.01** |
| 220 | **92.92 ± 0.00** | 88.08 ± 0.01 | 90.83 ± 0.01 | 92.17 ± 0.01 | 87.37 ± 0.02 | 92.44 ± 0.01 | 88.43 ± 0.01 | 91.70 ± 0.01 | 92.53 ± 0.01 | **92.99 ± 0.01** |
| 320 | **94.88 ± 0.00** | 90.99 ± 0.01 | 93.95 ± 0.00 | 94.67 ± 0.00 | 90.90 ± 0.02 | 94.30 ± 0.00 | 93.03 ± 0.00 | 94.30 ± 0.00 | 94.68 ± 0.00 | **94.93 ± 0.00** |
| 420 | **96.01 ± 0.00** | 92.45 ± 0.01 | 95.35 ± 0.00 | 95.87 ± 0.00 | 92.84 ± 0.02 | 95.43 ± 0.00 | 95.22 ± 0.00 | 95.82 ± 0.00 | 95.64 ± 0.01 | **96.09 ± 0.00** |
| 520 | **96.50 ± 0.00** | 93.51 ± 0.01 | 96.00 ± 0.00 | 96.34 ± 0.00 | 93.70 ± 0.01 | 96.13 ± 0.00 | 95.80 ± 0.00 | 96.46 ± 0.00 | 96.33 ± 0.00 | **96.64 ± 0.00** |
| 620 | **96.96 ± 0.00** | 94.22 ± 0.00 | 96.78 ± 0.00 | 96.93 ± 0.00 | 94.77 ± 0.01 | 96.63 ± 0.00 | 96.80 ± 0.00 | 96.94 ± 0.00 | 96.82 ± 0.00 | **97.11 ± 0.00** |
| 720 | **97.33 ± 0.00** | 94.78 ± 0.00 | 97.12 ± 0.00 | 97.15 ± 0.00 | 95.21 ± 0.01 | 96.98 ± 0.00 | 96.95 ± 0.00 | **97.36 ± 0.00** | 97.02 ± 0.00 | 97.28 ± 0.00 |
| 820 | **97.58 ± 0.00** | 95.18 ± 0.00 | 97.41 ± 0.00 | 97.46 ± 0.00 | 95.38 ± 0.01 | 97.06 ± 0.00 | 97.31 ± 0.00 | **97.56 ± 0.00** | 97.46 ± 0.00 | 97.49 ± 0.00 |
| 920 | **97.77 ± 0.00** | 95.44 ± 0.00 | **97.69 ± 0.00** | 97.68 ± 0.00 | 95.97 ± 0.00 | 97.44 ± 0.00 | 97.51 ± 0.00 | 97.63 ± 0.00 | 97.61 ± 0.00 | 97.65 ± 0.00 |
| 1,020 | **97.95 ± 0.00** | 95.88 ± 0.00 | 97.88 ± 0.00 | 97.84 ± 0.00 | 95.84 ± 0.01 | 97.63 ± 0.00 | 97.77 ± 0.00 | 97.75 ± 0.00 | 97.78 ± 0.00 | **97.97 ± 0.00** |
| **CIFAR10** | | | | | | | | | | |
| 1,400 | **50.30 ± 0.00** | 49.16 ± 0.01 | 48.96 ± 0.01 | 49.35 ± 0.01 | 47.44 ± 0.01 | 49.46 ± 0.01 | 49.51 ± 0.01 | 49.93 ± 0.01 | **50.23 ± 0.01** | 49.48 ± 0.00 |
| 2,600 | **57.01 ± 0.01** | 56.04 ± 0.01 | 55.35 ± 0.01 | 56.51 ± 0.01 | 52.95 ± 0.01 | 56.09 ± 0.01 | 55.53 ± 0.01 | 56.39 ± 0.01 | **56.74 ± 0.01** | 56.35 ± 0.01 |
| 3,800 | **61.30 ± 0.01** | 59.99 ± 0.00 | 59.68 ± 0.01 | 60.49 ± 0.00 | 56.30 ± 0.01 | 59.99 ± 0.00 | 59.43 ± 0.01 | 60.34 ± 0.01 | **60.54 ± 0.01** | 60.39 ± 0.01 |
| 5,000 | **64.09 ± 0.00** | 62.69 ± 0.01 | 62.60 ± 0.00 | 62.81 ± 0.01 | 58.70 ± 0.00 | 62.73 ± 0.01 | 62.31 ± 0.00 | 63.32 ± 0.00 | **63.93 ± 0.01** | 63.00 ± 0.00 |
| 6,200 | **65.79 ± 0.00** | 64.56 ± 0.01 | 64.42 ± 0.00 | 64.97 ± 0.00 | 61.08 ± 0.01 | 64.38 ± 0.01 | 64.00 ± 0.01 | 64.99 ± 0.01 | 65.30 ± 0.00 | **65.32 ± 0.00** |
| 7,400 | **67.28 ± 0.01** | 65.87 ± 0.00 | 66.29 ± 0.00 | 66.75 ± 0.01 | 62.84 ± 0.01 | 66.28 ± 0.01 | 66.12 ± 0.01 | 66.78 ± 0.01 | **67.11 ± 0.01** | 66.82 ± 0.01 |
| 8,600 | **68.83 ± 0.00** | 67.50 ± 0.01 | 67.77 ± 0.01 | 67.87 ± 0.01 | 63.50 ± 0.01 | 67.60 ± 0.00 | 67.94 ± 0.01 | 68.11 ± 0.01 | **68.42 ± 0.00** | 67.99 ± 0.00 |
| 9,800 | **70.10 ± 0.00** | 68.51 ± 0.01 | 68.62 ± 0.00 | 69.13 ± 0.01 | 64.44 ± 0.01 | 68.45 ± 0.01 | 68.82 ± 0.01 | 69.23 ± 0.01 | 69.08 ± 0.01 | **69.42 ± 0.01** |
| **SVHN** | | | | | | | | | | |
| 1,400 | **79.02 ± 0.01** | 77.22 ± 0.01 | 76.25 ± 0.01 | 77.94 ± 0.01 | 75.84 ± 0.01 | 77.56 ± 0.01 | **78.42 ± 0.01** | 77.49 ± 0.01 | 78.31 ± 0.01 | 77.91 ± 0.00 |
| 2,600 | **83.76 ± 0.00** | 81.55 ± 0.01 | 82.33 ± 0.00 | **83.61 ± 0.01** | 80.62 ± 0.01 | 82.55 ± 0.01 | 83.27 ± 0.01 | 83.23 ± 0.01 | 83.49 ± 0.00 | 83.13 ± 0.00 |
| 3,800 | **86.08 ± 0.00** | 83.61 ± 0.00 | 85.33 ± 0.00 | 85.28 ± 0.01 | 81.95 ± 0.01 | 84.59 ± 0.00 | **86.00 ± 0.00** | 85.85 ± 0.01 | 85.68 ± 0.00 | 85.54 ± 0.00 |
| 5,000 | **87.58 ± 0.00** | 85.05 ± 0.00 | 86.78 ± 0.00 | 87.11 ± 0.00 | 83.12 ± 0.00 | 85.92 ± 0.00 | **87.49 ± 0.00** | 87.23 ± 0.00 | 87.32 ± 0.01 | 87.49 ± 0.00 |
| 6,200 | **88.64 ± 0.00** | 85.98 ± 0.00 | 88.09 ± 0.00 | 88.11 ± 0.00 | 84.21 ± 0.00 | 86.89 ± 0.00 | **88.91 ± 0.00** | 88.30 ± 0.00 | 88.38 ± 0.00 | 88.61 ± 0.01 |
| 7,400 | **89.56 ± 0.00** | 86.59 ± 0.00 | 88.84 ± 0.00 | 89.03 ± 0.00 | 84.73 ± 0.00 | 87.57 ± 0.00 | **89.61 ± 0.00** | 89.31 ± 0.01 | 89.20 ± 0.00 | 89.29 ± 0.00 |
| 8,600 | **90.04 ± 0.00** | 87.28 ± 0.00 | 89.65 ± 0.00 | 89.62 ± 0.00 | 85.25 ± 0.00 | 88.22 ± 0.00 | **90.34 ± 0.00** | 90.04 ± 0.00 | 89.90 ± 0.00 | **90.07 ± 0.00** |
| 9,800 | **90.61 ± 0.00** | 87.72 ± 0.00 | 90.16 ± 0.00 | 90.45 ± 0.00 | 86.13 ± 0.00 | 88.80 ± 0.00 | **90.77 ± 0.00** | 90.32 ± 0.00 | 90.10 ± 0.00 | 90.52 ± 0.00 |

Continued on next page

Table 5 – Continued from previous page

| $\mathcal{L}$ | LDM-S | Rand | Entropy | Margin | Coreset | ProbCov | DBAL | ENS | BADGE | BAIT |
|---|---|---|---|---|---|---|---|---|---|---|
| **CIFAR100** | | | | | | | | | | |
| 7,000 | **31.85 ± 0.00** | 31.28 ± 0.01 | 30.74 ± 0.01 | 31.18 ± 0.01 | 31.46 ± 0.01 | **31.76 ± 0.01** | 30.80 ± 0.01 | 31.09 ± 0.01 | 30.89 ± 0.00 | |
| 9,000 | **37.61 ± 0.00** | 36.94 ± 0.01 | 36.30 ± 0.00 | 36.83 ± 0.02 | 37.60 ± 0.01 | **37.68 ± 0.01** | 36.32 ± 0.01 | 36.42 ± 0.00 | 36.69 ± 0.01 | |
| 11,000 | 40.88 ± 0.01 | 40.40 ± 0.02 | 39.91 ± 0.01 | 40.33 ± 0.01 | **41.33 ± 0.01** | **41.43 ± 0.01** | 40.24 ± 0.01 | 40.04 ± 0.00 | 40.34 ± 0.01 | - |
| 13,000 | 43.86 ± 0.01 | 43.13 ± 0.00 | 43.16 ± 0.01 | 43.51 ± 0.01 | **44.15 ± 0.00** | **44.02 ± 0.01** | 43.32 ± 0.01 | 42.80 ± 0.01 | 43.44 ± 0.01 | |
| 15,000 | **46.59 ± 0.01** | 45.77 ± 0.01 | 45.77 ± 0.01 | 46.10 ± 0.00 | **46.58 ± 0.01** | 46.55 ± 0.01 | 46.11 ± 0.01 | 45.53 ± 0.01 | 46.20 ± 0.01 | |
| 17,000 | **48.81 ± 0.01** | 47.75 ± 0.01 | 47.86 ± 0.01 | 48.36 ± 0.00 | **48.69 ± 0.01** | 48.24 ± 0.01 | 48.23 ± 0.01 | 47.58 ± 0.01 | 48.46 ± 0.00 | |
| 19,000 | **49.41 ± 0.01** | 48.35 ± 0.01 | 48.66 ± 0.00 | 48.96 ± 0.01 | **49.28 ± 0.01** | 48.82 ± 0.01 | 48.99 ± 0.01 | 48.52 ± 0.01 | 49.20 ± 0.01 | |
| 21,000 | **50.94 ± 0.00** | 49.87 ± 0.01 | 50.49 ± 0.01 | 50.56 ± 0.01 | **50.71 ± 0.01** | 50.34 ± 0.00 | 50.60 ± 0.01 | 50.10 ± 0.01 | 50.54 ± 0.01 | |
| 23,000 | **52.48 ± 0.00** | 51.29 ± 0.01 | 52.10 ± 0.01 | 52.04 ± 0.01 | 51.93 ± 0.00 | 52.02 ± 0.00 | **52.29 ± 0.00** | 51.74 ± 0.01 | 52.06 ± 0.01 | |
| 25,000 | **56.00 ± 0.00** | 54.51 ± 0.01 | **55.73 ± 0.01** | 55.52 ± 0.00 | 55.15 ± 0.01 | 55.70 ± 0.01 | 55.72 ± 0.01 | 55.03 ± 0.01 | 55.63 ± 0.01 | |
| **Tiny ImageNet** | | | | | | | | | | |
| 15,000 | **23.97 ± 0.00** | 23.72 ± 0.01 | 23.13 ± 0.00 | 23.80 ± 0.01 | 23.87 ± 0.01 | **24.06 ± 0.00** | 23.71 ± 0.01 | 23.23 ± 0.01 | 23.70 ± 0.01 | |
| 20,000 | **27.76 ± 0.00** | 27.65 ± 0.00 | 26.72 ± 0.00 | 27.39 ± 0.01 | 27.67 ± 0.01 | **28.22 ± 0.01** | 27.70 ± 0.01 | 27.29 ± 0.01 | 27.65 ± 0.00 | |
| 25,000 | **31.51 ± 0.01** | 31.22 ± 0.00 | 30.30 ± 0.01 | 30.90 ± 0.00 | 31.18 ± 0.00 | **31.64 ± 0.01** | 31.30 ± 0.01 | 30.82 ± 0.01 | 31.21 ± 0.00 | |
| 30,000 | **34.39 ± 0.01** | 34.23 ± 0.01 | 33.38 ± 0.01 | 33.72 ± 0.00 | 33.85 ± 0.01 | **34.54 ± 0.01** | 34.27 ± 0.01 | 33.73 ± 0.00 | 34.20 ± 0.00 | - |
| 35,000 | **37.43 ± 0.01** | 36.77 ± 0.01 | 36.16 ± 0.00 | 36.78 ± 0.00 | 36.38 ± 0.01 | 36.79 ± 0.01 | 36.88 ± 0.00 | 36.11 ± 0.01 | **36.88 ± 0.01** | |
| 40,000 | **39.39 ± 0.01** | 38.59 ± 0.01 | 38.34 ± 0.00 | **39.51 ± 0.01** | 38.36 ± 0.01 | 39.02 ± 0.02 | 39.08 ± 0.01 | 38.43 ± 0.01 | 38.90 ± 0.01 | |
| 45,000 | **41.32 ± 0.01** | 40.25 ± 0.02 | 40.04 ± 0.01 | **41.50 ± 0.00** | 39.97 ± 0.01 | 40.73 ± 0.00 | 40.92 ± 0.02 | 40.73 ± 0.01 | 40.70 ± 0.01 | |
| 50,000 | 42.65 ± 0.01 | 41.73 ± 0.01 | 41.40 ± 0.01 | **43.01 ± 0.01** | 41.16 ± 0.02 | 42.70 ± 0.01 | 42.59 ± 0.02 | **42.90 ± 0.01** | 42.01 ± 0.01 | |
| **FOOD101** | | | | | | | | | | |
| 9,000 | **26.32 ± 0.00** | 25.86 ± 0.01 | 25.42 ± 0.00 | 25.63 ± 0.01 | **26.49 ± 0.01** | 25.91 ± 0.00 | 26.06 ± 0.01 | 25.54 ± 0.00 | 25.55 ± 0.01 | |
| 12,000 | **30.83 ± 0.01** | 29.89 ± 0.01 | 29.10 ± 0.00 | 29.77 ± 0.01 | **31.03 ± 0.00** | 30.40 ± 0.00 | 30.44 ± 0.01 | 29.56 ± 0.01 | 29.83 ± 0.01 | |
| 15,000 | 35.08 ± 0.01 | 34.19 ± 0.00 | 33.01 ± 0.01 | 34.31 ± 0.01 | **35.40 ± 0.01** | 34.70 ± 0.01 | **35.12 ± 0.01** | 33.71 ± 0.01 | 34.32 ± 0.01 | |
| 18,000 | **38.50 ± 0.01** | 37.15 ± 0.01 | 36.18 ± 0.01 | 37.68 ± 0.01 | **38.62 ± 0.01** | 38.00 ± 0.01 | 38.48 ± 0.00 | 36.82 ± 0.01 | 37.88 ± 0.01 | - |
| 21,000 | **42.26 ± 0.01** | 40.74 ± 0.00 | 39.71 ± 0.01 | 41.53 ± 0.01 | **42.37 ± 0.01** | 41.60 ± 0.01 | 42.16 ± 0.01 | 40.66 ± 0.01 | 41.69 ± 0.01 | |
| 24,000 | **45.46 ± 0.01** | 43.81 ± 0.01 | 43.24 ± 0.01 | 44.48 ± 0.01 | **45.24 ± 0.01** | 44.58 ± 0.01 | 45.20 ± 0.01 | 43.93 ± 0.01 | 45.17 ± 0.01 | |
| 27,000 | **48.88 ± 0.00** | 47.37 ± 0.01 | 46.64 ± 0.01 | 47.75 ± 0.01 | **48.70 ± 0.01** | 47.67 ± 0.00 | 48.69 ± 0.01 | 47.49 ± 0.00 | 48.39 ± 0.01 | |
| 30,000 | **51.40 ± 0.00** | 50.24 ± 0.00 | 49.67 ± 0.01 | 50.20 ± 0.00 | 51.22 ± 0.00 | 49.72 ± 0.01 | **51.52 ± 0.01** | 50.12 ± 0.01 | 50.93 ± 0.01 | |
| **ImageNet** | | | | | | | | | | |
| 192,180 | **41.99 ± 0.00** | 41.45 ± 0.01 | 41.25 ± 0.00 | 41.62 ± 0.01 | 41.27 ± 0.00 | 41.90 ± 0.01 | | **41.92 ± 0.00** | 41.90 ± 0.01 | |
| 256,240 | **46.62 ± 0.00** | 45.90 ± 0.00 | 45.58 ± 0.01 | 46.01 ± 0.01 | 45.41 ± 0.00 | 46.33 ± 0.00 | - | 46.49 ± 0.01 | **46.52 ± 0.01** | - |
| 320,300 | **50.14 ± 0.00** | 49.08 ± 0.01 | 48.86 ± 0.01 | 49.26 ± 0.01 | 48.38 ± 0.01 | 49.33 ± 0.01 | | 49.72 ± 0.01 | **49.88 ± 0.01** | |
| 384,360 | **53.53 ± 0.00** | 52.11 ± 0.00 | 52.18 ± 0.00 | 52.16 ± 0.01 | 50.96 ± 0.00 | 51.86 ± 0.00 | | 52.46 ± 0.00 | **52.95 ± 0.01** | |

