# OpenReview forum: "Querying Easily Flip-flopped Samples for Deep Active Learning"
_ICLR.cc/2024/Conference — ICLR 2024 poster_

### Official Review · Reviewer_L6Gr · 2023-10-30

**Soundness:** 4 excellent
**Presentation:** 4 excellent
**Contribution:** 3 good
**Rating:** 6
**Confidence:** 3

**Summary:**

This paper proposes a new active learning technique based on novel selection strategy of unlabeled examples. Its main contributions are the following:

— A new measure of closeness to the decision boundary the authors call the 'least disagree metric' (LDM). This is a metric inspired by the disagree metric of [1],  to play a role akin to "margin-score" or "entropy" in the standard active-learning algorithms like "margin" etc. As the authors mention, conceptually, a sample with a small LDM indicates that its prediction can be easily flip-flopped even by a small perturbation in the predictor.

— An estimator of LDM that is provably asymptotically consistent under mild assumptions, and a simple Bayesian-perspective-inspired algorithm to empirically evaluate such an estimator. (LDM is intractable to compute in most cases so the authors propose.)

— An LDM-based active learning algorithm (LDM-S) that, besides using LDM as the "scoring function" for selecting unlabeled examples",  it also makes sure that there's "diversity" in the selected batch of unlabeled examples to be labeled. In particular, diversity is ensured via a modification of the k-means++ seeding algorithm and  without introducing additional hyperparameters. Finally, The authors compare their algorithm with several SOTA active learning techniques and show that it typically performs better.

[1]  Theory of Disagreement-Based Active Learning. Foundations and Trends® in Machine Learning, 7(2-3):131–309, 2014, Steve Hanneke.

**Strengths:**

— Well-written paper

— Novel, principled approach to active-learning.

— A method of incorporating diversity in the sample selection without adding hyperparameters.

— Extensive experimental evaluation.

**Weaknesses:**

— The proposed approach provides somewhat mild benefits compared to already existing approaches (although it is consistently the best or the second best approach in each scenario, and the best according to metrics that consider the average performance among all datasets.)

—The datasets considered in this paper are somewhat small-scale, and so it's not clear to me whether the proposed approach is suitable for large-scale applications. For example, in Table 2 where the authors present the running time per dataset, Imagenet — which is the largest dataset considered in this paper, is missing.

**Questions:**

Could the authors provide the mean running time of each algorithm considered for the Imagenet dataset?

---

> ### Author Response · Authors · 2023-11-19
>
> We thank the reviewer for an overall positive outlook of our paper as well as insightful questions. Let us clarify the points raised in the review:
>
>
> > The proposed approach provides somewhat mild benefits compared to already existing approaches (although it is consistently the best or the second best approach in each scenario, and the best according to metrics that consider the average performance among all datasets.)
>
> We first emphasize that, as shown by our extensive experiments, our algorithm consistently outperforms or is at least at par with, prior state-of-the-art algorithms over various scenarios (varying datasets, networks, etc). This characteristic is very important as it shows how strong and safe our algorithm is, i.e., how robust our algorithm is to the scenario considered.
>
> We also note that our algorithm *appears* to provide mild benefits compared to existing approaches because it is compared with the best-performing algorithm for each dataset. Our algorithm provides significantly improved performance when comparing each algorithm across all datasets, which is arguably a fairer comparison. For example, our algorithm significantly outperforms OML#6, OML#44135, CIFAR10, SVHN, TinyImageNet, and FOOD101 compared to BADGE on all datasets. In particular, there is a large performance difference of about 2% between the two algorithms on OML#6 and OML#44135.
>
>
> > The datasets considered in this paper are somewhat small-scale, and so it's not clear to me whether the proposed approach is suitable for large-scale applications. For example, in Table 2 where the authors present the running time per dataset, Imagenet — which is the largest dataset considered in this paper, is missing.
>
> > Could the authors provide the mean running time of each algorithm considered for the Imagenet dataset?
>
> For ImageNet, we have used different hardware (CPU/GPU) for each algorithm due to time and resource constraints, so a fair comparison between the algorithms is impossible. Still, to answer your pointed-out weakness, one step of acquisition time (min) is reported.
> The results are as follows.
>
> | Algorithm  | time (min) |
> | -------- | ------- |
> |Training |	4,572.1, |
> |LDM-S |		     62.8, |
> |Entropy |	       5.5, |
> |Margin | 	       3.6, |
> |Coreset | 	     96.6, |
> |ProbCov | 	     17.0, |
> |ENS | 		     18.1, |
> |BADGE | 	1,389.9.|
>
> If we only compare the acquisition time of a single step, our algorithm seems to take longer than Entropy or Margin. Still, the difference is insignificant if we consider the training time of the model, which was three days. In the case of BADGE, the runtime is proportional to the square of the number of features and classes, so the runtime increases significantly on ImageNet compared to other datasets, which is in contrast to our algorithm, which is comparable to other algorithms. ENS has a short acquisition time, but this is because only the time for calculating the variation ratio is measured. If the whole training time of ensemble networks is considered, the runtime is about 15 days.

---

> > ### Comment · Reviewer_L6Gr · 2023-11-19
> > **Reply**
> >
> > Thank you for the additional experiments and explanations! I will keep my the score as is — but I am still an advocate of the paper being accepted.

---

> ### Author Response · Authors · 2023-11-20
> **Thank you for the review**
>
> Thank you for your constructive feedback that helped us further improve our paper and for a positive outlook. We are grateful to the reviewer for advocating our paper, and we will ensure that all the feedback and comments are incorporated into our final manuscript.

---

### Official Review · Reviewer_ixMJ · 2023-10-30

**Soundness:** 3 good
**Presentation:** 3 good
**Contribution:** 2 fair
**Rating:** 6
**Confidence:** 4

**Summary:**

This paper proposes to use LDM estimator as heuristics to query samples for active learning. The estimator is proven to be asymptotically consistent under mild assumptions. Two approaches based on LDM, i.e., naive approach and LDM-S, have been considered for batch active learning.

**Strengths:**

(1) The paper maintains a high-quality presentation. The measure, proof of asymptotical consistency, and algorithms are clearly presented.
(2) Extensive experiments on 3 openml datasets and 6 benchmark image datasets.

**Weaknesses:**

(1) Many baseline models are not considered in the paper's experiment, e.g., SAAL, Cluster-Margin, Similar, and [4].

(2) Computational cost analysis is lacking, including LDM estimation cost and LDM-S. What is the relationship to batch size, ensemble size, M, etc? The computational cost seems to be comparable to BADGE which has squared complexity to batch size.

(3) The authors provide no analysis of the relatedness of LDM to active learner performance in the paper setting.

(4) Careful discussion of limitations is lacking in the paper. There are many deep active learning algorithms coming out each year, each with its own pros and cons. A careful discussion of advantages and limitations will be very helpful to the community. One possible drawback is the need to use ensembles which takes more cost compared to methods that only use one model like BADGE and CoreSet.

(5) The seeding strategy seems detached from the novel estimator. Many strategies, e.g., [1], [2], and [4], have been published to extend to batches. It is better to compare these strategies based on LDM and show if the performance improvement is consistent.

references:

[1] Kim, Yoon-Yeong, Youngjae Cho, JoonHo Jang, Byeonghu Na, Yeongmin Kim, Kyungwoo Song, Wanmo Kang, and Il-chul Moon. "SAAL: Sharpness-Aware Active Learning." (2023).

[2] Citovsky, Gui, Giulia DeSalvo, Claudio Gentile, Lazaros Karydas, Anand Rajagopalan, Afshin Rostamizadeh, and Sanjiv Kumar. "Batch active learning at scale." Advances in Neural Information Processing Systems 34 (2021): 11933-11944.

[3] Kothawade, Suraj, Nathan Beck, Krishnateja Killamsetty, and Rishabh Iyer. "Similar: Submodular information measures based active learning in realistic scenarios." Advances in Neural Information Processing Systems 34 (2021): 18685-18697.

[4] Kirsch, Andreas, Sebastian Farquhar, Parmida Atighehchian, Andrew Jesson, Frédéric Branchaud-Charron, and Yarin Gal. "Stochastic Batch Acquisition: A Simple Baseline for Deep Active Learning." Transactions on Machine Learning Research (2023).

**Questions:**

(1) Could the authors provide numerical or illustrative examples to show that LDM-based algorithm can be more effective in deep active learning? The authors provided the asymptotical analysis of the estimator but no analysis is provided for the relatedness of LDM to the final performance. In this case, it is better to provide some illustrative examples to show the effectiveness of LDM-based algorithms.

(2) I am very interested to see the performance of active learning models with MC-dropout. MC-dropout is more efficient compared ensemble method.

(3) Could the authors explain why they do not use more advanced posterior sampling method like [1] and [2]

references:

[1] Zhang, Ruqi, Chunyuan Li, Jianyi Zhang, Changyou Chen, and Andrew Gordon Wilson. "Cyclical stochastic gradient MCMC for Bayesian deep learning." arXiv preprint arXiv:1902.03932 (2019).

[2] Chen, Tianqi, Emily Fox, and Carlos Guestrin. "Stochastic gradient hamiltonian monte carlo." In International conference on machine learning, pp. 1683-1691. PMLR, 2014.

---

> ### Author Response · Authors · 2023-11-19
>
> We would like to thank the reviewer for providing several constructive comments. Let us clarify the points raised in the review:
>
> > Many baseline models are not considered in the paper's experiment, e.g., SAAL, Cluster-Margin, Similar, and [4].
>
> We focused on uncertainty and diversity-baed baselines, which have been receiving lots of attention in recent years and which are much more related (and thus comparable) to our work. We emphasize that we have already considered quite a lot of standard baseline models to the best of our knowledge at the time of submission.
>
> We thank the reviewer for pointing out additional baselines compared to which we could further show the effectiveness of our approach. Out of the suggested algorithms, we have added BatchBALD (Figure 13) and Cluster-Margin (Figure 14) results in Appendix F.3, showing that, indeed, our LDM-S outperforms both algorithms. We will add the results for the other suggested algorithms in the final draft (we are currently implementing those algorithms).
>
>
> > Computational cost analysis is lacking, including LDM estimation cost and LDM-S. What is the relationship to batch size, ensemble size, M, etc? The computational cost seems to be comparable to BADGE which has squared complexity to batch size.
>
> First, we would like to emphasize that in our paper, “ensemble size” is the number of sampled hypotheses in the LDM estimation. In our paper, we do not perform the usual ensembling where multiple models must be trained with new initializations; instead, we replace this with sampling near a single trained model.
>
> We have already included the computational cost analysis for LDM evaluation in Figure 10 of Appendix F.1; this cost is directly related to the number of Monte-Carlo samples $M$ and the number of sampled hypotheses $N$. The runtime for LDM evaluation is almost linearly proportional to the stop condition $s$ and is linearly proportional to $M$ and $N$.
>
> Let us now clarify the dependency of the computational cost on the batch size. Our LDM-S has two stages: evaluating the LDM of each sample for the entire pool dataset (which only requires a single pass at the beginning) via Algorithm 1 and selecting samples for querying. The first stage is not affected by the batch size, but the runtime for the second stage increases linearly with the batch size since it requires selecting a batch size of samples. BADGE uses a similar seeding method; thus, its runtime increases linearly with the batch size. We would also like to clarify that BADGE has squared complexity to the number of features and classes, not the batch size.
>
>
> > The authors provide no analysis of the relatedness of LDM to active learner performance in the paper setting.
>
> > Could the authors provide numerical or illustrative examples to show that LDM-based algorithm can be more effective in deep active learning? The authors provided the asymptotical analysis of the estimator but no analysis is provided for the relatedness of LDM to the final performance. In this case, it is better to provide some illustrative examples to show the effectiveness of LDM-based algorithms.
>
> To show the effectiveness of LDM, in Appendix F.1, we provide an ablation study by replacing LDM with other uncertainty measures, such as Entropy, Margin…etc, combined with our seeding algorithm. There, it is clear that out of all the considered uncertainty measures, LDM is the most effective in the context of active learning. Please also refer to Fig. 2 in the main text for illustrative examples of the effectiveness of LDM.

---

> ### Author Response · Authors · 2023-11-19
>
> > Careful discussion of limitations is lacking in the paper. There are many deep active learning algorithms coming out each year, each with its own pros and cons. A careful discussion of advantages and limitations will be very helpful to the community. One possible drawback is the need to use ensembles which takes more cost compared to methods that only use one model like BADGE and CoreSet.
>
> Again, we would like to emphasize that in our paper, “ensemble size” is the number of sampled hypotheses in the LDM estimation. In our paper, we do not perform the usual ensembling where multiple models must be trained with multiple (random) initializations; instead, we replace this with sampling near a single trained model.
>
> Thus, the drawback regarding the cost, as suggested by the reviewer, is not accurate, as we only train one model as well, similar to BADGE and CoreSet; this is clear when we compare the runtime of the algorithms in Table 2 in the main text.
>
> Still, we agree with the reviewer that a careful discussion of the limitations of our framework is indeed lacking in our paper, and we would like to thank the reviewer for pointing this out. We have included another paragraph in the Conclusion section that discusses the potential limitations of our work. One limitation that is also an important future direction is to obtain a rigorous sample complexity guarantee for our LDM-S algorithm in the context of active learning; we have included this in our revised Conclusion section.
>
>
> > The seeding strategy seems detached from the novel estimator. Many strategies, e.g., [1], [2], and [4], have been published to extend to batches. It is better to compare these strategies based on LDM and show if the performance improvement is consistent.
>
> Thank you for pointing out these recent works. First, [1] uses k-means++ seeding to introduce diversity to the acquisition, similar to our approach, along with a novel acquisition function; replacing that with LDM is precisely our current algorithm.
>
> At first glance, Cluster Margin [2] is a viable option for LDM to be incorporated into, with LDM replacing margin score. However, this introduces an additional hyperparameter, margin batch size $k_m$, and the number of clusters $r$ among others, which requires an intricate tuning. This goes against our goal of keeping our proposed algorithm simple and intuitive. Also, during the preliminary phase of our research, we tried a similar approach in which we first sampled samples with high LDM (more uncertain) and then performed k-means clustering. Here, we observed that the tuning of the additional parameters relied heavily on the particular problem instance, and the overall performance was sensitive to such hyperparameters.
>
> [4], which is a very recent work, proposes several alternate batch acquisition strategies, such as score-based and rank-based strategies. Indeed, it would be interesting to see how our LDM-based approach would combine with the alternate strategies, and we will definitely look into this in our future work. We would like to emphasize that our main novelty lies in proposing LDM, a new uncertainty measure, which provides very good performance in the active learning context.
>
>
> > I am very interested to see the performance of active learning models with MC-dropout. MC-dropout is more efficient compared ensemble method.
>
> Again, we would like to emphasize that in our paper, “ensemble size” is the number of sampled hypotheses in the LDM estimation. In our paper, we do not perform the usual ensembling where multiple models must be trained; rather, we replace this with sampling near a single trained model.
>
> In Table 2, we compare our approach to the MC-dropout-based approach (DBAL [9]) has about the same runtime as ours, yet the number of hypotheses for MC-dropout is 100 while our approach uses much more (for instance, for MNIST + SCNN, with stop condition $s = 10$, our approach uses roughly $10^4$ hypotheses). It should also be noted that with similar time, our approach consistently outperforms, or is at least at par with, DBAL over all tasks. In this respect, although somewhat similar, MC-dropout requires a bit more computation time than ours, as it requires going through all the neuron connections in the given network for dropout for each forward pass, in contrast to ours, which only involves sampling. In other words, given the same computation cost, our approach could yield more hypotheses than MC dropout; of course, the precise comparison would be different depending on the network.

---

> > ### Author Response · Authors · 2023-11-19
> >
> > > Could the authors explain why they do not use more advanced posterior sampling method like [5] and [6]
> >
> > We did not pursue these directions as such advanced posterior sampling methods involve additional hyperparameters (e.g., initial step size, number of cycles, the proportion of exploration phase, in [5]) and additional computational cost/complexity (e.g., computation of diffusion matrix, friction coefficient in [6]). We have used a simple Gaussian sampling scheme for practical purposes in deep active learning. Indeed, empirically, we have shown that even this simple sampling scheme is sufficient for our purpose in deep active learning. We would like to add that SGD is still considered one of the standard and *scalable* Bayesian inference methods for deep neural networks [7,8]. Still, we agree that making/adapting advanced *simple* posterior sampling to our framework would be a fruitful future direction, and we have included the relevant discussions in the revised conclusion section.
> >
> >
> >
> > **References:**
> >
> > [1] Kim, Yoon-Yeong, Youngjae Cho, JoonHo Jang, Byeonghu Na, Yeongmin Kim, Kyungwoo Song, Wanmo Kang, and Il-chul Moon. "SAAL: Sharpness-Aware Active Learning." (2023).
> >
> > [2] Citovsky, Gui, Giulia DeSalvo, Claudio Gentile, Lazaros Karydas, Anand Rajagopalan, Afshin Rostamizadeh, and Sanjiv Kumar. "Batch active learning at scale." Advances in Neural Information Processing Systems 34 (2021): 11933-11944.
> >
> > [3] Kothawade, Suraj, Nathan Beck, Krishnateja Killamsetty, and Rishabh Iyer. "Similar: Submodular information measures based active learning in realistic scenarios." Advances in Neural Information Processing Systems 34 (2021): 18685-18697.
> >
> > [4] Kirsch, Andreas, Sebastian Farquhar, Parmida Atighehchian, Andrew Jesson, Frédéric Branchaud-Charron, and Yarin Gal. "Stochastic Batch Acquisition: A Simple Baseline for Deep Active Learning." Transactions on Machine Learning Research (2023).
> >
> > [5] Zhang, Ruqi, Chunyuan Li, Jianyi Zhang, Changyou Chen, and Andrew Gordon Wilson. "Cyclical stochastic gradient MCMC for Bayesian deep learning." arXiv preprint arXiv:1902.03932 (2019).
> >
> > [6] Chen, Tianqi, Emily Fox, and Carlos Guestrin. "Stochastic gradient hamiltonian monte carlo." In International conference on machine learning, pp. 1683-1691. PMLR, 2014.
> >
> > [7] Mingard, Chris, Valle-Perez, Guillermo, Skalse, Joar, and Louis, Ard A. “Is SGD a Bayesian sampler? Well, almost.” In Journal of Machine Learning Research. 2021.
> >
> > [8] Kim, Balhae, Choi, Jungwon, Lee, Seanie, Lee, Yoonho, Ha, Jung-Woo, and Lee, Juho. “On Divergence Measures for Bayesian Pseudocoresets.” In NeurIPS 2022.
> >
> > [9] Gal, Yarin, Islam, Riashat, and Ghahramani, Zoubin. “Deep Bayesian Active Learning with Image Data.” In ICML 2017.

---

> > > ### Comment · Reviewer_ixMJ · 2023-11-22
> > > **Thank you for the detailed response**
> > >
> > > I have read the response. Some of the questions are resolved. Some concerns including weakness 5 and question 1 remain. The score is raised accordingly.

---

> > > > ### Author Response · Authors · 2023-11-22
> > > >
> > > > First, we would like to thank the reviewer for providing detailed comments and feedback that helped improve our paper. We also thank the reviewer for revising the score. Still, we would like to try to address some of the remaining concerns that the reviewer has mentioned:
> > > >
> > > >
> > > > ***Weakness 5.*** To address the reviewer's concern, we plan to provide the results of Cluster Margin [2] combined with LDM for at least the MNIST dataset by tomorrow (before the discussion period ends). We will report this result and compare it to our approach, the k-means++ seeding algorithm combined with LDM. We will also add the results to our manuscript accordingly.
> > > >
> > > >
> > > > ***Question 1.*** Let us summarize the experiments and ablations supporting our claim that our LDM-based algorithm is effective in deep active learning. First, the overall superior (or at least on par) performance of our LDM-based algorithm indirectly shows the desired effectiveness. We also provided several ablations (for 2D binary classification, MNIST, CIFAR10, CIFAR100)  showing that when LDM is replaced with other uncertainty measures, such as Entropy or Margin, there is a noticeable performance degradation (Section 4.1, Appendix G.1). We believe that this ablation directly shows the effectiveness of our LDM as an effective uncertainty measure.
> > > >
> > > > We will continue to consider ways (experiments, ablations, theories, etc.) to better show our LDM-based algorithm's effectiveness in deep active learning. If the reviewer has any suggestions or comments, we would be more than happy to incorporate them into our final manuscript and try to answer them before the discussion period ends.

---

> > > > > ### Author Response · Authors · 2023-11-23
> > > > >
> > > > > We would like to thank the reviewer for patiently waiting. In the second revision, we've included the results of the Cluster-Margin + LDM in Fig. 12 of Appendix F.2.
> > > > >
> > > > > There, it is observed that Cluster-Margin combined with our LDM (which is done by replacing the margin score with LDM score) underperforms compared to our k-means++ seeding-based approach; actually, it can be seen that the performance degrades with Cluster-Margin. As we've added in our discussion in Appendix F.2, we believe this is because Cluster-Margin *strongly* pursues batch diversity, diminishing the effect of LDM as an uncertainty measure. Specifically, Cluster-Margin considers samples of varying LDM scores from the beginning (a somewhat large portion of them are thus not so useful). In contrast, our algorithm significantly weights the samples with small LDM, biasing our samples towards more uncertain samples (and thus more useful).
> > > > >
> > > > > We believe that this additional experiment, as suggested by the reviewer, further shows the effectiveness of our suggested LDM.
> > > > >
> > > > > Thank you for your helpful suggestion, and we hope that this further answers your concern.

---

> ### Comment · Area_Chair_ASPk · 2023-11-21
>
> Dear reviewer ixMJ, please review and respond the the authors' responses to your initial review.
>
> Thank you for your critical contribution to this conference.

---

### Official Review · Reviewer_o58y · 2023-10-31

**Soundness:** 3 good
**Presentation:** 3 good
**Contribution:** 3 good
**Rating:** 8
**Confidence:** 3

**Summary:**

Framed in the field of active learning, this paper presents LDM (least disagree metric), a novel concept to quantify the distance between an instance and the decision boundary of a classifier. Along with the theoretical definition of LDM, the authors provide an asimptotically consistent estimator of LDM as well as a practical algorithm to calculate it. Based on this notion of LDM, they define a new acquisition procedure in active learning, by favoring low values of LDMs and enforcing diversity. Empirical evaluation shows that the proposed approach is competitive or superior against other state-of-the-art active learning methods in various datasets and with different architectures.

**Strengths:**

* The quality of the exposition is high in general. The concepts and ideas are presented by combining an specific and accurate definition along with an intuition on their meaning. It is easy to follow the flow of the paper, and sections are well organized in a natural manner.

* The experimental evaluation is a comprehensive one, including a wide range of baselines, datasets and architectures. Results are analyzed in a rigorous way, including statistical tests and popular active learning metrics such as the performance profile (Dolan-More plots).

* The proposed metric LDM is clearly explained, and intuition is provided on how it reflects the distance to the decision boundary. It is well motivated and theoretically sounded. The example provided after Definition 1 is clarifying and helps understanding LDM definition.

* LDM estimator, denoted as $L_{N,M}$, has a theoretical underpinning. Under some assumptions, $L_{N,M}$ is shown to converge in probability to the true value L when M scales logarithmically with respect to N.

* 2D binary classification case is thoroughly investigated. Proposition 1 and Proposition 2 in the appendix theoretically confirm the intuition behind the choices made in this work.

* Additional interesting experiments are included in the appendix. In particular:
    * An effort to study the hyperparameters effect (number of MC samples, batch size, LDM stop condition and σ interval) is made.
    * To favour diversity, the LDM estimator is corrected using a popular weighting strategy
in active learning. The importance of this correction is addressed in the ablation study.

**Weaknesses:**

* I think there exists an important gap between the theoretical description of LDM (its definition in Section 2.1 and its estimator in Section 2.2) and how it is empirically evaluated (Section 2.3). In Section 2.3, the "motivation" paragraph includes several sentences to justify the procedure that the authors are going to follow to empirically evaluate LDM, but these sentences are just somewhat "generic"/"loose", and there is no guarantee that hypothesis in Section 2.2 are satisfied. Taking this into account, I wonder whether Section 2.2 is a core component of the contribution, or could be moved to the appendix, leaving room in the main text for other information that may be more central to the contribution (e.g. a more detailed description of the experiments or further empirical evaluations, which are currently in the appendix).

* Even if Section 2.2 is moved to the appendix, I think that assumptions 1, 2 and 3 should be further motivated. The reader who is unfamiliar with mathematical concepts could get lost: what is a Polish space, and why is it necessary? Why is it necessary that ρ is Lipschitz? How restrictive is assumption 3? There are no references to other works where these assumptions are made. If they are common, these references should be provided. If not, authors should motivate and discuss them.

* I also wonder whether the idea of "sampling close to the decision boundary" is the best way to go in active learning. Samples close to the decision boundary may not be informative if all the uncertainty that they present is of _aleatoric_ nature (inherent to the data, i.e. it cannot be reduced by further sampling training data). I think that more subtle distinctions on the types of uncertainty to be considered in active learning should be analyzed. My intuition is that this might be related to the finding that using LDM alone did not work properly, and a (somewhat ad-hoc) procedure to encourage diversity had to be introduced.

* I think that more empirical evaluations should be moved to the main text (specially if theoretical details are moved to appendix, as suggested in the first point above). Right now, the experimental setting is not clearly described in the main text (it is deferred to Appendix C), and several ablation studies that could be interesting are "lost" in the appendix.

* In some occasions, the statements made by the authors may lead to confusion. For example, in section 2.3, it is said that “...in Appendix B.2, we show that $E_w[ρ_M(h, g)]$ is monotone increasing in $σ^2$.”. One could think that this is shown in general. However, the proof only applies to the 2D binary classification case.

* Related to the first point above, some aspects of theory are disconnected from the implementation:
    * To implement $L_{N,M}$ the authors propose to sample near the learned hypothesis using standard parameter perturbation techniques. As authors state, this sampling scheme would need to satisfy Assumption 3. However, it is not explained why this specific form
of sampling assures that the hypothesis spaces $\mathcal{H}_n$ satisfy Assumption 3.
    * In the 2D binary classification experiment, the performance of LDM-S, entropy, and random based sampling procedures are investigated. As authors state, true LDM is measurable in the 2D binary classification scenario. Thus, a study could be carried out on the error made when approximating L using $L_{N,M}$. Verification of bounds and rates of
convergence could be carried out.

**Questions:**

Other questions/comments:

* The effect of each hyperparameter is studied independently. Nothing is said about how they
affect each other. Does the choice of one hyperparameter (number of MC samples, batch size, LDM stop condition
and σ interval) affect each other?

* Which metric/score is used to quantify performance in section 4.2? Accuracy is mentioned in
the appendix, but it is not entirely clear to me.

* x should be present in the input in algorithm 1, right?

* Perhaps, Assumption 3 is over-complicated: $\sup_{\epsilon\in(0,1)} \lim_N α(N, \epsilon)= 0$ amounts to saying
that $\lim_N α(N, \epsilon)$ exists and is 0 when $\epsilon\in (0, 1)$.

* I think $f$ in Theorem 1 should be $\alpha$?

---

> ### Author Response · Authors · 2023-11-19
>
> We would like to thank the reviewer for providing several insightful and constructive comments and pointing out several typos we have missed. Let us clarify the points raised in the review:
>
>
> > I think there exists an important gap between the theoretical description of LDM (its definition in Section 2.1 and its estimator in Section 2.2) and how it is empirically evaluated (Section 2.3). In Section 2.3, the "motivation" paragraph includes several sentences to justify the procedure that the authors are going to follow to empirically evaluate LDM, but these sentences are just somewhat "generic"/"loose", and there is no guarantee that hypothesis in Section 2.2 are satisfied. Taking this into account, I wonder whether Section 2.2 is a core component of the contribution, or could be moved to the appendix, leaving room in the main text for other information that may be more central to the contribution (e.g. a more detailed description of the experiments or further empirical evaluations, which are currently in the appendix).
>
> >Related to the first point above, some aspects of theory are disconnected from the implementation:
> > - To implement $L_{N, M}$ the authors propose to sample near the learned hypothesis using standard parameter perturbation techniques. As authors state, this sampling scheme would need to satisfy Assumption 3. However, it is not explained why this specific form of sampling assures that the hypothesis spaces $\mathcal{H}_n$ satisfy Assumption 3.
> > - In the 2D binary classification experiment, the performance of LDM-S, entropy, and random based sampling procedures are investigated. As authors state, true LDM is measurable in the 2D binary classification scenario. Thus, a study could be carried out on the error made when approximating L using $L_{N, M}$. Verification of bounds and rates of convergence could be carried out.
>
>
> First, we would like to thank the reviewer for providing very insightful and constructive comments and questions that would help us improve our paper significantly. Indeed, as the reviewer has correctly stated, in the original draft, we did not theoretically justify our Algorithm 1 for evaluating LDM, thus seemingly creating a gap between Section 2.2 and 2.3.
>
> However, we would like to emphasize that the current Algorithm 1 is *inspired* by our theoretical guarantees, and thus, we believe that Section 2.2 is a core contribution of our work. While proving the asymptotic consistency of the LDM estimator, we realized that Assumption 3 (coverage assumption) was required. Naturally, we thought that by sampling a sufficiently large, finite collection of $N$ hypotheses $\mathcal{H}_N$ with a distribution whose support spans the entire space, e.g., Gaussian distribution over $\mathbb{R}^d$ and is centered around $g$, we could “cover” the hypothesis that yields the true LDM with high probability, as $N \rightarrow \infty$. Conversely, if such a true hypothesis cannot be covered, we cannot hope to obtain a consistent estimator of LDM.
>
> We agree with the reviewer that this still does not give a solid reason why the proposed sampling scheme should satisfy Assumption 3. But, we would like to remark that, as briefly mentioned in Section 2.3, directly verifying Assumption 3 for a very general scenario (e.g., deep neural network + multiclass classification + complex data distribution) is often impossible, as it is challenging to compute the true LDM nor identify the hypothesis that (approximately) attains the true LDM. On a slightly different note, we have shown in Appendix F.1 that for realistic deep AL scenarios, our algorithm shows a solid trend of convergence in the estimated LDM, which suggests that Assumption 3 is being satisfied.
>
> We have also empirically shown that for the toy binary classification with linear classifiers in which the true LDM and the optimal hypothesis attaining the optimal LDM are known (see Section 2.1), the empirical LDM output from our algorithm converges to the true LDM. As the reviewer has suggested, we could prove that Assumption 3 can be attained in the same setting. We have included this in Appendix B.3. This shows that Assumption 3 is not unrealistic and is not an assumption for the considered toy example. Again, we thank the reviewer for this helpful suggestion.
>
> Lastly, we agree with the reviewer that some of the experiments in the appendix should be moved to the main text. We have done so in the revised manuscript; see the general response for the changes that we have made.

---

> > ### Author Response · Authors · 2023-11-19
> >
> > > Even if Section 2.2 is moved to the appendix, I think that assumptions 1, 2 and 3 should be further motivated. The reader who is unfamiliar with mathematical concepts could get lost: what is a Polish space, and why is it necessary? Why is it necessary that ρ is Lipschitz? How restrictive is assumption 3? There are no references to other works where these assumptions are made. If they are common, these references should be provided. If not, authors should motivate and discuss them.
> >
> > We thank the reviewer for pointing this out. Below, we motivate each of the assumptions, all of which have been added to the revised manuscript:
> >
> > Assumption 1. Why $\mathcal{H}$ should be Polish?
> > The reason for assuming that $\mathcal{H}$ is a Polish space is to avoid any complications that may arise from the space being not countable, especially as we consider a probability measure over $\mathcal{H}$; see Chapter 1.1 of [1], where it states that the usual measurability and other related properties may not hold for non-separable spaces  (e.g., Skorohod space). The Euclidean space, which comes up naturally by parametrizing $\mathcal{H}$ (e.g., neural networks), is Polish.
> >
> > [1] Aad W. van der Vaart and Jon A. Wellner. “Weak Convergence and Empirical Processes: With Applications to Statistics,” Springer. 1996.
> >
> >
> > Assumption 2. Why is it necessary for $\rho$ to be Lipschitz?
> > If $\rho$ is not Lipshitz, then the disagree metric may behave arbitrarily regardless of whether the hypotheses are “close” or not. This can occur in specific (arguably not so realistic) corner cases such as discrete data distribution (i.e., a mixture of Dirac measures).
> >
> > Assumption 3. How restrictive is this assumption?
> > Please see our answer to the previous question.
> >
> >
> > > I think that more empirical evaluations should be moved to the main text (especially if theoretical details are moved to the appendix, as suggested in the first point above). Right now, the experimental setting is not clearly described in the main text (it is deferred to Appendix C), and several ablation studies that could be interesting are "lost" in the appendix.
> >
> > We strongly agree with the reviewer that our paper would benefit significantly by moving additional experiments and experimental settings from the appendix to the main text. To make room, we moved the related work section to the appendix. We then revived the diversity ablation (described as “extraordinary” by reviewer 2mvn). Unfortunately, due to the strict page limit, we could not move the experimental details in Appendix C to the main text.
> >
> >
> > > I also wonder whether the idea of "sampling close to the decision boundary" is the best way to go in active learning. Samples close to the decision boundary may not be informative if all the uncertainty that they present is of aleatoric nature (inherent to the data, i.e. it cannot be reduced by further sampling training data). I think that more subtle distinctions on the types of uncertainty to be considered in active learning should be analyzed. My intuition is that this might be related to the finding that using LDM alone did not work properly, and a (somewhat ad-hoc) procedure to encourage diversity had to be introduced.
> >
> > Thank you for providing an insightful comment regarding the nature of uncertainty in active learning. Indeed, as the reviewer has correctly pointed out, our LDM-based approach focuses on tackling epistemic uncertainty by considering the prediction's disagreement and does not explicitly take the inherent aleatoric uncertainty (e.g., confidence) of the prediction into account; the distinction between epistemic and aleatoric uncertainty in the context of active learning was studied in [2]. Also, the reviewer’s intuition is correct. The randomness from our seeding algorithm seems to allow it to consider samples with small epistemic uncertainty but possibly large aleatoric uncertainty. Also, such randomness can help to cover the entire distribution of attributes and labels.
> >
> > [2] Nguyen, Vu-Linh, Mohammad Hossein Shaker, and Eyke Hüllermeier. "How to measure uncertainty in uncertainty sampling for active learning." Machine Learning 111.1 (2022): 89-122.
> >
> >
> >
> > > In some occasions, the statements made by the authors may lead to confusion. For example, in section 2.3, it is said that “...in Appendix B.2, we show that $\mathbb{E}_\bm{w} [\rho_{_M} (h, g)] is monotone increasing in $\sigma^2$.” One could think that this is shown in general. However, the proof only applies to the 2D binary classification case.
> >
> > We apologize for the confusion. Indeed, our theoretical analysis in Appendix C.2 regarding the intuitions behind our design choices only holds for the 2D binary classification case; we have clarified this by making the sentences clearer.
> >
> > However, we also emphasize that we have experimentally verified (in Appendix B) that the statements are roughly true for more general classification tasks. We have included this additional discussion in the main text.

---

> ### Author Response · Authors · 2023-11-19
>
> > The effect of each hyperparameter is studied independently. Nothing is said about how they affect each other. Does the choice of one hyperparameter (number of MC samples, batch size, LDM stop condition and σ interval) affect each other?
>
> Thank you for asking this question. The number of MC samples, $M$, affects runtime and the estimation quality of the LDM, but it has little effect on other hyperparameters or performance, i.e., $M$ is independent of other hyperparameters. Batch size is also irrelevant as it is given by the task rather than being tuned individually in the algorithm by the learner.
>
> The stop condition, $s$, and the $\sigma$ interval can indirectly affect each other. If the $\sigma$ interval is narrow, the estimation quality of LDM would be high with smaller $s$, although the number of intervals to be considered would also have to increase. If the $\sigma$ interval is wide, then to retain the estimation quality, $s$ also needs to be increased, but the number of intervals wouldn’t be as high as before. In our setting, we set the $\sigma$ interval sufficiently narrow for using a small $s$ to reduce runtime, which we have verified empirically to be the case when we grid-searched for the correct hyperparameters.
>
>
>
> > Which metric/score is used to quantify performance in section 4.2? Accuracy is mentioned in the appendix, but it is not entirely clear to me.
>
> In Section 4.3 (Main Experiments), we use two performance metrics. We use test accuracy (%) for Table 1 and AUC for the Dolan-More curve (Fig. 3). We have included the exact expressions for the metrics for Fig. 3 in Appendix E. We have clarified this further in our revised manuscript.
>
>
> > x should be present in the input in algorithm 1, right?
>
> Thank you for pointing this out. Yes, it should be part of the input. We have fixed this in our revised manuscript.
>
>
> > Perhaps, Assumption 3 is over-complicated: $\sup_{\epsilon \in (0, 1)} \lim_{N} \alpha(N, \epsilon) = 0$ amounts to saying that $\lim_{N} \alpha(N, \epsilon)$ exists and is 0 when $\epsilon \in (0, 1)$.
>
> That is true. We wanted to state that $\lim_N \alpha(N, \varepsilon) = 0$ for any $\epsilon \in (0, 1)$ in a compact way. But we are happy to rephrase the assumption to be in a simpler term, as the reviewer has suggested.
>
>
> > I think $f$ in Theorem 1 should be $\alpha$?
>
> Indeed, $f$ should be $\alpha$. Thank you for pointing out this typo. We have fixed it.

---

> ### Comment · Reviewer_o58y · 2023-11-21
> **Reply**
>
> Thanks to the authors for their detailed response, which has addressed a good amount of my concerns. Thus, I am raising my score to 8.

---

> > ### Author Response · Authors · 2023-11-22
> >
> > Thank you again for your detailed and insightful comments and feedback. We are delighted that our response has answered a good amount of the reviewer's concerns, and we thank the reviewer for revising the score. We will ensure all the discussions are incorporated into our final manuscript.

---

### Official Review · Reviewer_2mvn · 2023-11-01

**Soundness:** 4 excellent
**Presentation:** 4 excellent
**Contribution:** 3 good
**Rating:** 10
**Confidence:** 4

**Summary:**

This paper presents LDM-S, an active learning method based on selecting samples close to the decision boundary. The paper starts by formulating a Least Disagree Metric (LDM) function, and then proceeds to implement it in a Monte Carlo fashion. Necessary assumptions and theorems are proven as needed. Over 5 repeats, the method is compared with other active learning methods and a t-test is used to check significance.

**Strengths:**

* At least in some parts of the paper, the flow is very good, and theorems and conclusions naturally lead to the next part.
* The evidence for diversity in LDM-based active learning, as discussed in Appendix E.3, is extraordinary. Through an intuitive first example and then a real-world example (MNIST), the motivation is very clear. If possible, this should definitely be in the main paper to give readers better intuitions.
* Ablation studies are presented whenever necessary to justify choices made, for example, for modifying the k-means++ seeding algorithm.

**Weaknesses:**

* Minor typo: Page 1, last paragraph: flips-flopped --> flip-flopped
* Section 2.2 is rather rushed in its presentation, and details are not expanded on. For example, it is unclear why $\mathcal{H}$ needs to be a Polish space (i.e., why second countability is necessary, for instance). In Theorem 1, $f$ is undefined. In Assumption 3, it seems that the phrase, "that is monotone decreasing in the first argument" refers to $\alpha$, but that is not fully clear.
* Similarly, some other details are presented with no clear rationale. For example, in (6), why is p(x) squared?
* In Figure 2a, LDM sometimes samples on or very close to the decision boundary. Please change the color of the LDM crosses so they are more apparent.

**Questions:**

* Why does $\mathcal{H}$ need to be a Polish space as opposed to a more general metric space?
* In Algorithm 2, when you compute $L_x$ via Algorithm 1, could you clarify the parameters passed? My understanding is that you pass it a hypothesis parameterized by $v$, number of samples $m$, and a "small" $s$ (as said in Sec. 2.3), but I'm unclear what $\{ \sigma^2_k \}$ you pass.

---

> ### Author Response · Authors · 2023-11-19
>
> We would like to thank the reviewer for highlighting the strengths of our paper and providing several constructive comments. Let us clarify the points raised in the review:
>
>
> > Minor typo: Page 1, last paragraph: flips-flopped --> flip-flopped
>
> Thank you for pointing out this typo. We have fixed this in our revised manuscript.
>
>
> > Section 2.2 is rather rushed in its presentation, and details are not expanded on. For example, it is unclear why $\mathcal{H}$  needs to be a Polish space (i.e., why second countability is necessary, for instance). In Theorem 1, $f$  is undefined. In Assumption 3, it seems that the phrase, "that is monotone decreasing in the first argument" refers to $\alpha$ but that is not fully clear.
>
> > Why does $\mathcal{H}$ need to be a Polish space as opposed to a more general metric space?
>
> We thank the reviewer for pointing this out. We have revised Section 2.2 to be more well-motivated and have expanded on the missing details.
>
> Thank you for pointing out the typos. Indeed, $f$ should be $\alpha$, and with a closer look, we realized that the “monotone …” part of the assumption isn’t required. We have removed that part, and we would like to thank the reviewer for helping us discover this.
>
> The reason for assuming that $\mathcal{H}$ is a Polish space is to avoid any complications that may arise from the space being not countable, especially as we consider a probability measure over $\mathcal{H}$; see Chapter 1.1 of [1], where it states that the usual measurability and other related properties may not hold for non-separable spaces (e.g., Skorohod space). The Euclidean space, which comes up naturally by parametrizing $\mathcal{H}$ (e.g., neural networks), is Polish.
>
>
> [1] Aad W. van der Vaart and Jon A. Wellner. “Weak Convergence and Empirical Processes: With Applications to Statistics,” Springer. 1996.
>
> > Similarly, some other details are presented with no clear rationale. For example, in (6), why is p(x) squared?
>
> Our paper incorporates diversity by using the $k$-means++ seeding algorithm, which has been proposed in an influential prior work [1] for initializing the centers of clustering. Their seeding algorithm uses squared $p(x)$, i.e., squared distance, to minimize the potential function defined as $\phi = \sum_{x \in \mathcal{X}} \min_{c \in C} \Vert x - c \Vert^2$. Note that we only modified the distance measure from L2 to cosine distance between the features of the data points. In Appendix F.2, we have provided further ablation studies showing that the cosine distance outperforms L2 distance in active learning.
>
> Please let us know of any other confusing details; we will be happy to elaborate.
>
>
> [1] David Arthur and Sergei Vassilvitskii. K-Means++: The advantages of Careful Seeding. In Proceedings of the Eighteenth Annual ACM-SIAM Symposium on Discrete Algorithms, SODA ’07, 2007.
>
> > In Figure 2a, LDM sometimes samples on or very close to the decision boundary. Please change the color of the LDM crosses so they are more apparent.
>
> Thank you for your suggestion. We have revised the figure accordingly.
>
>
> > In Algorithm 2, when you compute $L_x$ via Algorithm 1, could you clarify the parameters passed? My understanding is that you pass it a hypothesis parameterized by $v$, number of samples $m$, and a "small" $s$ (as said in Sec. 2.3), but I'm unclear what $\sigma_k^2$ you pass.
>
> As described in Section 4.2, we set $s=10$, $M$ to be the same size as the pool size, and $\sigma_k = 10^{0.1k - 5}$ for $k \in {1, 2, \cdot, 51\}$. Figure 7 of Appendix C.2 shows the relationship between the disagree metric and $\sigma$. The $\sigma$ controls the $\rho$. To properly approximate LDM, we need to sample hypotheses with a wide range of $\rho$, and thus we need a wide range of $\sigma$. To efficiently cover a wide range of $\sigma$, we consider equal spacing in the log scale, where the spacing was chosen via a grid search.

---

> > ### Comment · Reviewer_2mvn · 2023-11-20
> >
> > The authors thoroughly addressed my concerns, and have updated their manuscript also taking other reviewers' comments into account. The new version is a significant improvement, with much more clear motivations and better references, and also moves the old Appendix E.3 to the main text.
> >
> > Based on the revised manuscript along with the authors' responses to all the reviewers, I would champion for strong acceptance, and have changed my score to 10/4. On top of the great presentation, the paper will be of interest to relevant communities in the ICLR audience.

---

> > > ### Author Response · Authors · 2023-11-20
> > > **Thank you for the review**
> > >
> > > Thank you for your constructive feedback and for revising the score. We are particularly grateful for their reviewer's overwhelmingly positive comments and enthusiasm for our paper. We will incorporate the suggested discussion and clarification into our final manuscript.

---

### Author Response · Authors · 2023-11-19
**General Response**

We thank the reviewers for taking the valuable time to read through the draft and giving constructive comments that would help us greatly improve our manuscript.

We are especially encouraged by the positive comments for the high quality of exposition and good flow (all reviewers), comprehensive experiments/ablations (all reviewers), clear motivations and novelty (reviewer 2mvn, o58y, L6Gr), and theoretical soundness of LDM (reviewer 2mvn, o58y, ixMJ).

We have carefully revised our manuscript by taking into account the reviewers’ comments, summarized here:
- Various typos that the reviewers have pointed out have been fixed
- Related work has been relegated to Appendix A.
- Ablation on the diversity has been moved to the main text (Section 4.2). Further discussions have been relegated to Appendix C.4, as mentioned in Section 4.2
- Section 2.2 and the Motivation paragraph of Section 2.3 have been reinforced with additional discussions regarding the assumptions and how the theoretical analyses motivated Alg. 1
- Additional theoretical result on Assumption 3 has been added in Appendix C.3
- Figure 11 (d-e), reporting the number of sampled hypotheses and runtime for evaluating LDM, has been added in Appendix F.1
- In Appendix F.3, we’ve added additional experimental results on BatchBALD and Cluster-Margin

(Please note that due to this rearrangement, the section alphabets of the Appendix and some of the subsection numbers have been shifted. In our response, we refer to the section numbers and appendix alphabets as in our revised manuscript, and we thank the reviewers in advance for your kind understanding.)

These updates have been highlighted in $\color{blue}\text{blue}$ for clarity.

We hope that our responses below and the revision address all of your concerns, and we are happy to address any more concerns or questions that the reviewers may have. We will continue to put in the best of our effort to provide satisfactory feedback throughout the discussion period.

Thanks,
Authors.

---

### Meta-Review · Area_Chair_ASPk · 2023-12-05

**Metareview:**

The authors provide a propose a novel active learning algorithm based on the concept of lease disagreement metric (LDM), provide an efficient (and provably consistent) estimator. The authors combine this metric with with a diversity encouraging sampling scheme, resulting in the LDM-S algorithm, which the authors evaluate via ablations and comparisons to 8 other active learning baselines across 9 datasets (with some additional experiments during the discussion phase). The empirical evaluation shows that the proposed method is on average better than other baseline with a comparable running time.

All reviewers appreciate the clarity of presentation (especially after revision), with strong intuition and theoretical justification, as well as the empirical evaluations (again, especially after some additions for the revision).

**Justification For Why Not Higher Score:**

The submission is clearly high quality, but not at the level of ground-breaking results or novel ideas that will affect an entire line of research. Thus, I think a poster (or arguably spotlight) is appropriate.

**Justification For Why Not Lower Score:**

All reviewers agree the paper should be accepted. The submission makes both theoretical and empirical contributions and overall presents a strong new active learning approach.

---

### Decision · Program_Chairs · 2024-01-16

Accept (poster)